# Different hotspot p53 mutants exert distinct phenotypes and predict outcome of colorectal cancer patients

Ori Hassin[1], Nishanth Belugali Nataraj[2], Michal Shreberk-Shaked[1], Yael Aylon[1], Rona Yaeger[3], Giulia Fontemaggi[4], Saptaparna Mukherjee[1], Martino Maddalena[1], Adi Avioz[1], Ortal Iancu[5], Giuseppe Mallel[6], Anat Gershoni[1], Inna Grosheva[7], Ester Feldmesser[8], Shifra Ben-Dor[8], Ofra Golani[8], Ayal Hendel[5], Giovanni Blandino[4], David Kelsen[3], Yosef Yarden[2] & Moshe Oren[1✉]

The TP53 gene is mutated in approximately 60% of all colorectal cancer (CRC) cases. Over 20% of all TP53-mutated CRC tumors carry missense mutations at position R175 or R273. Here we report that CRC tumors harboring R273 mutations are more prone to progress to metastatic disease, with decreased survival, than those with R175 mutations. We identify a distinct transcriptional signature orchestrated by p53R273H, implicating activation of oncogenic signaling pathways and predicting worse outcome. These features are shared also with the hotspot mutants p53R248Q and p53R248W. p53R273H selectively promotes rapid CRC cell spreading, migration, invasion and metastasis. The transcriptional output of p53R273H is associated with preferential binding to regulatory elements of R273 signature genes. Thus, different TP53 missense mutations contribute differently to cancer progression. Elucidation of the differential impact of distinct TP53 mutations on disease features may make TP53 mutational information more actionable, holding potential for better precision-based medicine.

[1] Department of Molecular Cell Biology, Weizmann Institute of Science, Rehovot, Israel. [2] Department of Biological Regulation, Weizmann Institute of Science, Rehovot, Israel. [3] Department of Medicine, Memorial Sloan Kettering Cancer Center, New York, NY, USA. [4] Oncogenomic and Epigenetic Unit, IRCCS Regina Elena National Cancer Institute, Rome, Italy. [5] The Institute for Advanced Materials and Nanotechnology, The Mina and Everard Goodman Faculty of Life Sciences, Bar-Ilan University, Ramat-Gan, Israel. [6] Pathology Department, Curesponse Ltd, Rehovot, Israel. [7] Department of Immunology, Weizmann Institute of Science, Rehovot, Israel. [8] Department of Life Sciences Core Facilities, Weizmann Institute of Science, Rehovot, Israel. ✉email: moshe.oren@weizmann.ac.il

The *TP53* gene, encoding the p53 tumor suppressor protein, is frequently mutated in many types of human cancer[1,2]. The most common type of *TP53* mutations are missense mutations, leading to a single amino acid substitution in an otherwise intact p53 protein. In addition, *TP53* nonsense and frameshift mutations, usually resulting in production of truncated p53 proteins, are also fairly common in cancer[3]. The common and arguably most important consequence of all these different types of mutations is the partial or complete loss of the tumor suppressor effects of the wild type (wt) p53 protein. Yet, there is growing evidence that missense *TP53* mutations may often also confer upon the mutant p53 (mutp53) proteins oncogenic gain-of-function (GOF) properties, which can actively contribute to cancer-related processes[4–9].

The spectrum of *TP53* missense mutations in human cancer comprises hundreds of different variants, although a small number of hotspot mutations are observed more frequently[10]. Broadly speaking, cancer-associated p53 missense mutant proteins can be divided into two main classes: (A) structural mutants, where the mutation causes misfolding of the protein and leads to a significant conformational alterations within p53's DNA binding domain (DBD), and (B) DNA contact mutants, where the overall structure of the DBD is only minimally perturbed, but the mutant protein loses its ability to engage in high-affinity sequence-specific interactions with p53 binding sites within the DNA[11,12]. Both mutp53 classes fail to activate canonical wtp53 target genes, but can modify the cell transcriptome through protein-protein interactions that involve a multitude of transcription factors and other DNA binding proteins[5,7].

While most of the studies on mutp53 have addressed features shared by all common mutants, there also is evidence for mutant-specific effects[5,13–17]. Notably, knock-in mice harboring different p53 mutations exhibit non-identical tumor phenotypes: $p53^{R270H/+}$ mice, corresponding to the human $p53^{R273H}$ DNA contact hotspot mutation, show increased incidence of carcinomas and B cell lymphomas compared to $p53^{+/-}$ mice, while $p53^{R172H/+}$ mice, corresponding to the human $p53^{R175H}$ structural hotspot mutation, frequently develop osteosarcomas[18]. However, the clinical implications of such mutant-specific differences remain largely unknown.

Colorectal cancer (CRC) is the 2nd most common cause of cancer-related deaths worldwide[19]. The malignant progression of CRC is driven largely by the sequential accumulation of genetic alterations, affecting both oncogenes and tumor suppressor genes[20,21]. Like other cancer types, CRC displays a wide spectrum of *TP53* mutations, which are observed in approximately 60% of all CRC tumors and are usually associated with the transition from large adenoma to invasive carcinoma[20].

In this study, we compare the impact of the two most common hotspot *TP53* mutations in CRC, $p53^{R273H}$ and $p53^{R175H}$. Interestingly, we find marked differences between the effects of these two mutants. Specifically, $p53^{R273H}$ but not $p53^{R175H}$ can orchestrate a unique transcriptional program, which drives oncogenic signaling pathways, leads to more aggressive disease, and is associated with significant differences in patient survival. Moreover, the hotspot mutations $p53^{R248Q}$ and $p53^{R248W}$ behave similarly to $p53^{R273H}$ in CRC. Better understanding of the distinct contributions of different *TP53* mutants might guide better CRC patient management and treatment decisions.

## Results

**p53 R273 mutants are associated with more aggressive colorectal tumors relative to R175 mutants**. Compared to most other cancers, in colorectal cancer (CRC) the relative representation of "hotspot" missense mutations among carriers of *TP53* mutations is particularly high. Specifically, missense mutations in the four most commonly mutated p53 residues (R175, R248, R273 and R282) comprise approximately 37% of all *TP53* mutations in this type of cancer (Supplementary Fig. 1a). In contrast, mutations in these four residues encompass only 17% of all *TP53* mutations in all other cancer types together. Although this might be simply due to the mutational signature of particular carcinogens, it might also suggest a more significant GOF effect of such missense mutations in CRC.

One obvious question is whether different hotspot mutations may exert different effects on disease features and patient outcome. To address this question, we set out to compare R175 structural mutations to R273 DNA contact mutations. Notably, these mutations together represent over 20% of all CRC tumors harboring *TP53* mutations, as compared to only approximately 10% in all other cancers (Fig. 1a). We analyzed clinical data from several patient cohorts, using the TCGA and ICGC open-source platforms as well as additional published datasets[22–24] (Supplementary Data 1). Remarkably, while R175 mutations are significantly more frequent than R273 mutations in early disease stages, the predominance of R175 mutations is abolished at later stages (Fig. 1b). This suggests that, relative to R175 mutations, R273 mutations might accelerate disease progression from early stages to advanced stages, involving cancer cell spreading to nearby lymph nodes (stage 3) and metastases to distant organs (stage 4).

Interestingly, when we analyzed the MSKCC CRC dataset, comprising 1134 cases of which ~90% were metastatic[25], we found that while both R175 and R273 mutants exhibited a similar percentage of liver, lung and lymph node first site metastases, R273 mutants were significantly more associated with tumors that metastasize first to less common sites such as brain, bone, pelvis, peritoneum and gynecological sites (Fig. 1c). Importantly, unlike liver and lung metastases, metastatic lesions in these sites are usually considered unresectable, and thus incurable. Indeed, many studies have linked the presence of metastases at those sites to worse survival[26–28]. Furthermore, R273 mutants were found to be significantly associated with multiple metastatic sites at the time of diagnosis of metastatic disease (Fig. 1d), further supporting the notion that R273 mutants selectively augment the metastatic capacity of CRC cancer cells. Importantly, R273 mutants were associated with significantly shorter disease-specific overall survival than R175 mutants (Fig. 1e), regardless of patient age, tumor location or presence of *KRAS* mutations (Fig. 1f and Supplementary Table 1). Interestingly, while R273 mutant tumors were associated with reduced survival of both male and female CRC patients (Supplementary Fig. 1b–c), the magnitude of the effect was greater in males (Supplementary Fig. 1c). Thus, the gender disparities in the impact of p53 status on cancer[29] might extend also to differences between individual p53 mutants.

To explore the possibility that R273 mutant tumors might be associated with a particular mutational landscape, which may account for the observed clinical effects, we compared the co-occurrence of the most common gene mutations in CRC with either R175 or R273 mutations. Notably, other than SMAD4 mutations which showed a mild co-occurrence with R273 mutations ($P = 0.02$), all other gene mutations were not differentially enriched in R273 mutated vs R175 mutated tumors (Supplementary Fig. 1d).

In sum, compared to R175 mutations, R273 mutations are preferentially associated with more advanced disease, higher rate of multiple and uncommon metastases, and shorter patient survival.

**$p53^{R273H}$ orchestrates a distinct transcriptional signature**. We next wished to elucidate the molecular mechanisms underpinning the differential impact of R273 vs R175 mutants in CRC, and to

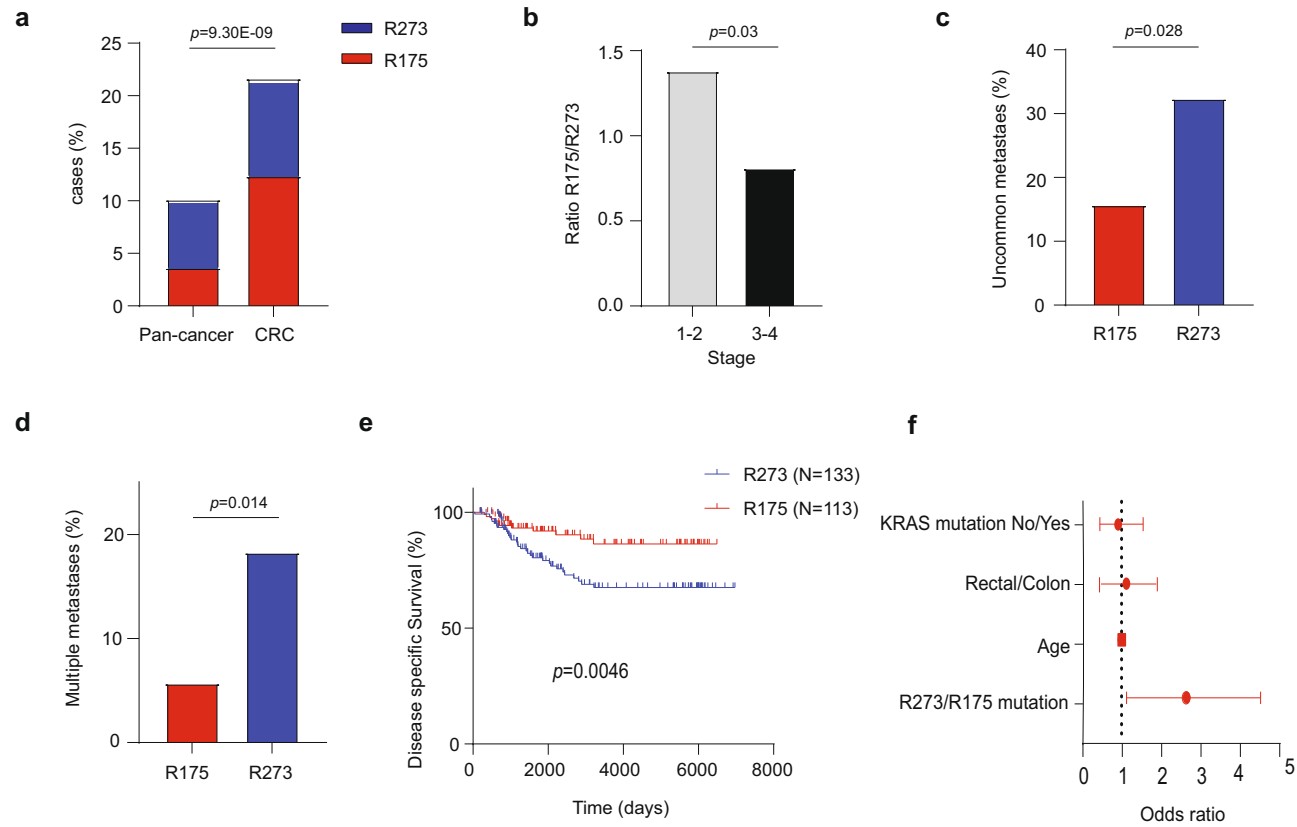

**Fig. 1 *TP53* R273 mutations in CRC are preferentially associated with more aggressive cancer features and shorter overall survival. a** Relative abundance of R175 and R273 *TP53* hotspot mutations in colorectal cancer (CRC, $n = 323$) versus all other cancers (Pan-cancer, $n = 3396$) in TCGA. Shown is the % of cases with each hotspot mutation out of all *TP53*-mutated cases. Two sided Fisher's exact test. **b** Ratio between the numbers of CRC cases with R175 mutations ($N = 132$) and R273 mutations ($N = 121$) in stage 1–2 and stage 3–4 disease. Two sided Fisher's exact test. **c** Percentage of cases of each mutation type with metastases at uncommon sites (brain, bone, pelvis, peritoneum and omentum) at presentation ($N = 66$ for R175 tumors and $N = 68$ for R273 tumors), in the MSKCC cohort. Two sided Fisher's exact test. **d** Percentage of cases of each mutation type ($N = 66$ for R175 tumors and $N = 68$ for R273 tumors) with multiple metastases (three or more) at presentation, in the MSKCC cohort. Two sided Fisher's exact test. **e** Disease specific overall survival of CRC patients with either R175 or R273 mutations. Compiled from TCGA COAD-READ and published data[24]. Log-rank test. **f** Multivariate Cox regression analysis for the impact of multiple variables on overall survival in the patient collection described in (**e**). Ovals represent hazard ratios, and error bars (horizontal lines) denote confidence intervals. Source data is provided as a Source Data file.

assess whether R273 mutations confer a true GOF. To that end, we utilized CRC-derived SW480 cells. SW480 is a microsatellite stable cell line, harboring APC and KRAS mutations; hence, it properly represents sporadic CRC. SW480 cells possess 3 copies of the *TP53* gene, each copy carrying the same two missense mutations: R273H and P309S[30]. SW480 cells depleted of their endogenous mutp53 by CRISPR/Cas9-mediated knockout (p53KO)[31] were stably transduced with either p53$^{R273H}$ or p53$^{R175H}$ (Fig. 2a). Western blot analysis confirmed comparable overexpression of both mutants (Fig. 2b). As mutp53 GOF often involves changes in the cell transcriptome, we next subjected the different SW480 cell pools to RNA sequencing (RNA-seq) analysis, using the MARS-seq protocol[32]. Clustering analysis revealed substantial differences between the transcriptome of the R273H cells and the parental p53KO cells (Fig. 2c). Surprisingly, overexpression of p53$^{R175H}$ had rather limited impact on the transcriptome of these cells (Supplementary Fig. 2a). By comparing the observed transcriptional profiles, we generated a gene signature comprising 140 genes upregulated by p53$^{R273H}$ relative to both p53$^{R175H}$ and p53KO cells. This gene signature was defined as the "R273 signature" (Fig. 2d and Supplementary Table 2).

To further validate our conclusions, we adopted an alternative approach wherein SW480 cells were stably transduced with shRNA directed against the 3' UTR of the *TP53* gene (shp53), followed by

stable overexpression of shRNA-resistant p53$^{R175H}$ or p53$^{R273H}$ (Fig. 2e). The resultant cell pools were subjected to MARS-seq analysis as above. Clustering analysis of the data confirmed that, also by this approach, p53$^{R273H}$ had a stronger effect on the SW480 cell transcriptome than p53$^{R175H}$ (Supplementary Fig. 2b). Importantly, gene set enrichment analysis (GSEA) confirmed that the "R273 signature", derived from the reconstituted p53KO cells, was selectively enriched upon p53$^{R273H}$ overexpression also in the shp53-based system, relative to the control shp53 cells (Fig. 2f) or the p53$^{R175H}$ overexpressors (Fig. 2g).

Last, since the above RNA-seq analyses were done with ectopically overexpressed p53 mutants, we quantified the relative expression of representative R273 signature genes by RT-qPCR analysis in control parental SW480 cells (expressing endogenous p53$^{R273H}$ and p53$^{P309S}$) and p53KO cells (Western blot in Supplementary Fig. 2c). As seen in Supplementary Fig. 2d, all tested genes were significantly downregulated in the knockout cells relative to the control parental cells, while being upregulated in the p53$^{R273H}$ overexpressors. Moreover, comparison by GSEA of our R273 signature to published RNA-seq data of SW480 cells before and after shRNA-mediated p53 knockdown[33] confirmed significantly higher expression of the R273 signature in the control cells (Supplementary Fig. 2e). Thus, p53$^{R273H}$ drives a distinct transcriptional program in SW480 cells.

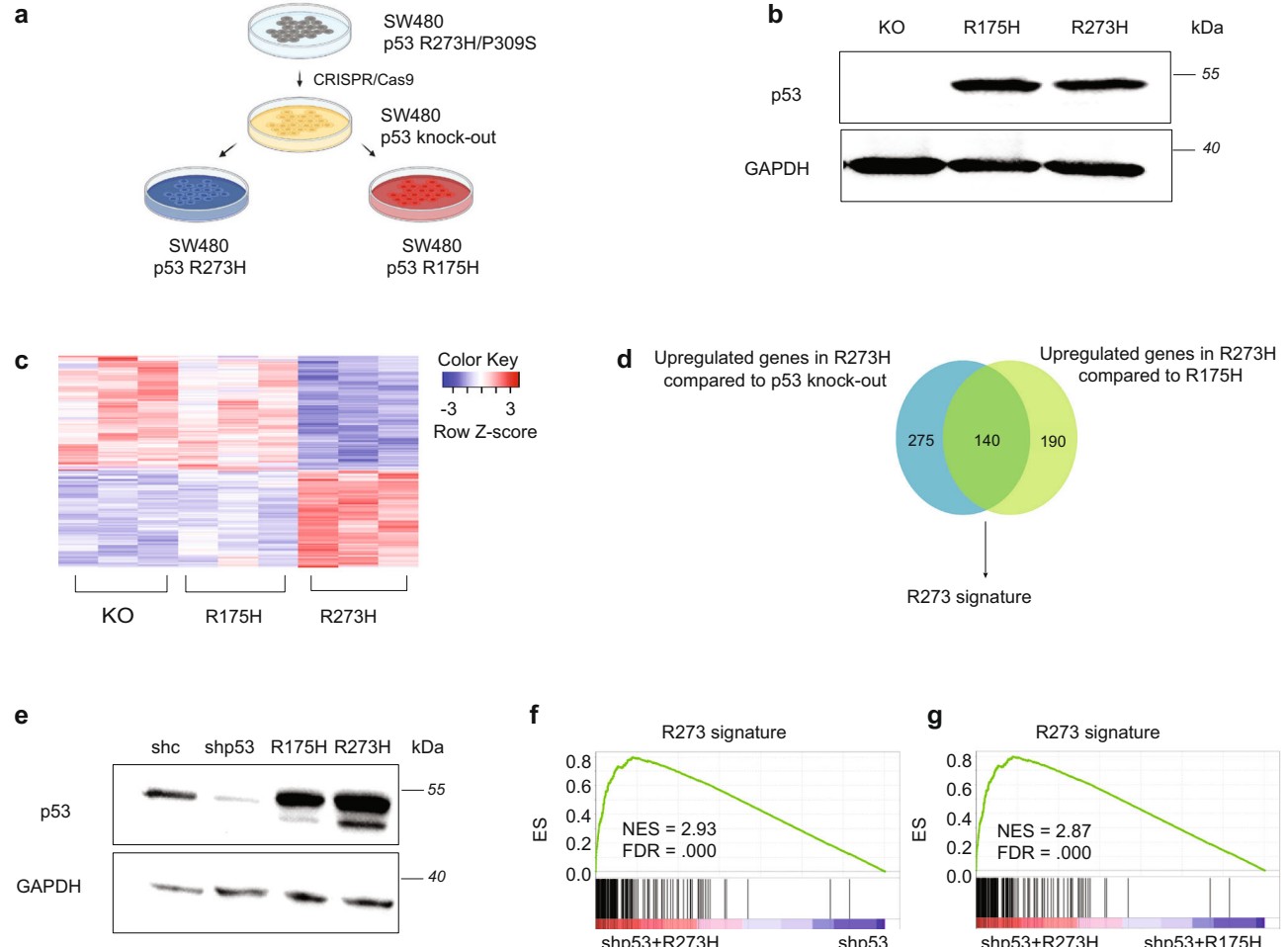

**Fig. 2 R273 mutants orchestrate a distinct transcriptional signature. a** SW480 cells in which the endogenous *TP53* genes (harboring R273H and P309S mutations) had been knocked out, were stably transduced with p53$^{R175H}$ or p53$^{R273H}$. **b** Western blot analysis of p53 in SW480 knockout (KO) cells before and after transduction of p53$^{R175H}$ or p53$^{R273H}$. n = 3. **c** SW480 *TP53* KO cells and their derivatives expressing p53$^{R175H}$ or p53$^{R273H}$ were subjected to RNA-seq analysis. Shown is a heatmap of genes differentially expressed (fold change > 1.5, pAdj < 0.05) in p53$^{R273H}$ overexpressing cells relative to p53 KO and p53$^{R175H}$ overexpressing cells, n = 3. **d** Venn diagram of upregulated genes (fold change > 1.5, pAdj < 0.1) in p53$^{R273H}$ overexpressors relative to p53 KO cells (blue circle) or p53$^{R175H}$ overexpressors (green circle). The 140 overlapping genes were defined as the 'R273 signature'. **e** Western blot analysis of p53 in SW480 cells stably transduced with shRNA directed against the 3' UTR of the *TP53* gene (shp53), followed by stable overexpression of shRNA-resistant p53$^{R175H}$ or p53$^{R273H}$. shc = SW480 cells transduced with control shRNA, to visualize the endogenous p53. n = 3. **f, g** Gene Set Enrichment Analysis (GSEA) of differentially expressed genes in shp53 cells reconstituted with p53$^{R273H}$ vs control shp53 cells or shp53 cells reconstituted with p53$^{R175H}$ (ranked by fold change), using the R273 signature as the tested gene set. ES = Enrichment score. Source data is provided as a Source Data file.

**R273 signature genes are selectively upregulated by p53$^{R273H}$ in CRC cells.** The differential transcriptional effects of p53$^{R273H}$ vs p53$^{R175H}$, shown in Fig. 2, were observed in mutp53 over-expressing cells. To determine whether such differential effects are also evident when the two p53 mutants are expressed endogenously, we employed RNP-mediated CRISPR/Cas9 gene editing to replace the endogenous wild type *TP53* genes of HCT116 CRC cells with either p53$^{R273H}$ or p53$^{R175H}$ (Fig. 3a). For each mutant, five independent clones were validated by DNA sequencing, and endogenous p53 expression was verified by Western blot analysis (Supplementary Fig. 3a). RNA from each clone was then sub-jected to RT-qPCR analysis, and values from all 5 clones expressing the same mutant were averaged. As expected, both the p53$^{R273H}$ and p53$^{R175H}$ clones showed significant down-regulation of p21 mRNA levels (Fig. 3b), consistent with loss of wild type p53 function. Importantly, compared to either parental HCT116 cells or CRISPR/Cas9 control cells, the p53$^{R273H}$ knock-in clones displayed significant upregulation of representative

R273 signature genes (Fig. 3c). In contrast, these genes were upregulated only mildly, or not at all, in the R175H knock-in cells (Fig. 3c).

In a complementary approach, we employed shRNA-mediated knockdown to compare the effect of mutp53 depletion in two CRC cell lines, one (HT-29) harboring endogenous p53$^{R273H}$ and the other (COGA-5) harboring p53$^{R175H}$. As seen in Fig. 3d–g, while knockdown of p53$^{R273H}$ in HT-29 cells significantly downregulated the expression of most of the tested R273 signature genes, knockdown of p53$^{R175H}$ failed to exert a similar effect. Hence, p53$^{R273H}$ selectively upregulates R273 signature genes also when expressed endogenously in CRC cells.

To further assess the generality of the R273 signature, we expressed p53$^{R273H}$ and p53$^{R175H}$ ectopically in two additional CRC-derived cell lines: RKO cells, depleted of their endogenous wtp53 (KO)[34], and COLO-205 cells, which endogenously express truncated p53 (Supplementary Fig. 3b, d). Reassuringly, RT-

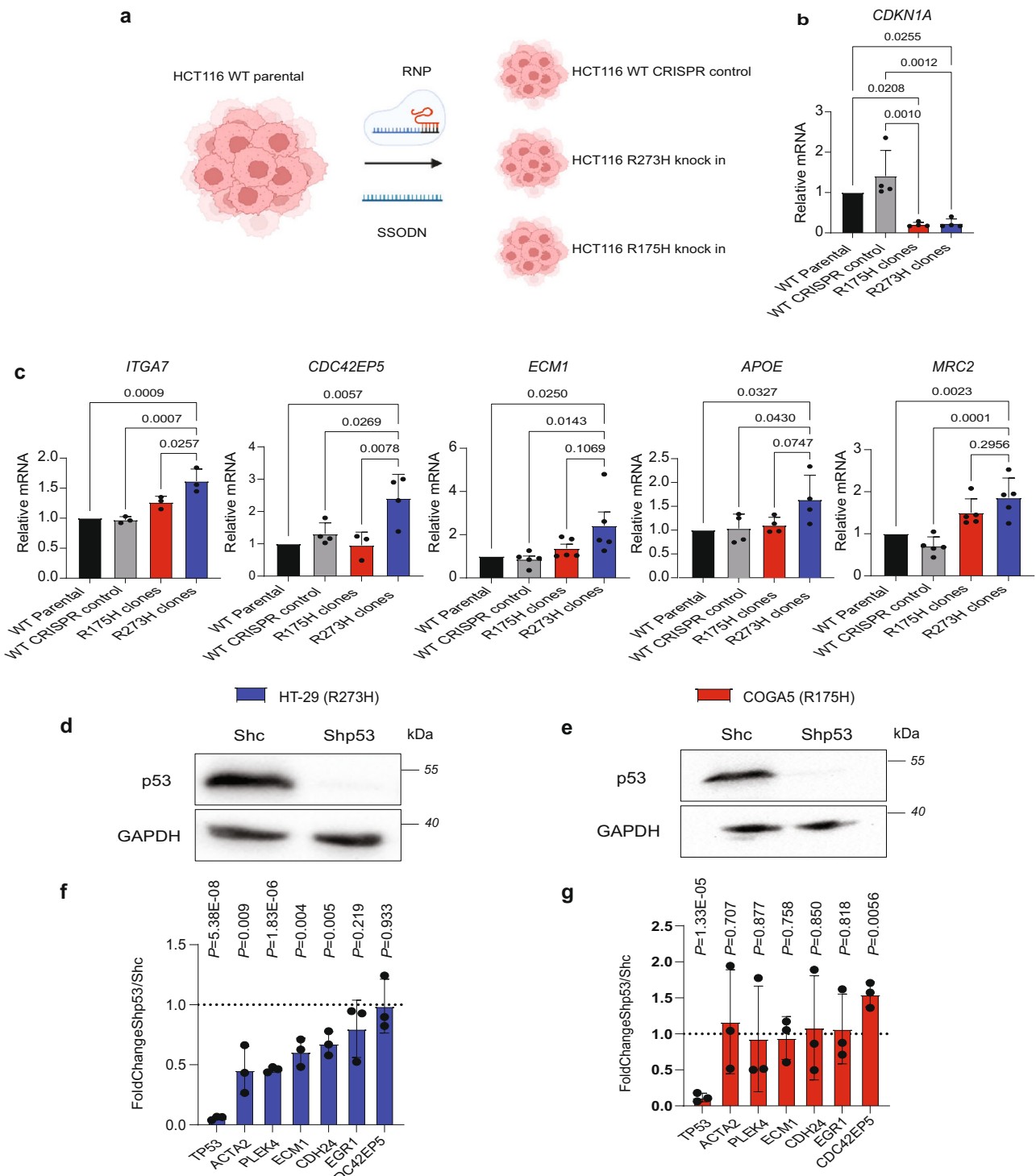

**Fig. 3 R273 signature genes are selectively upregulated by p53$^{R273H}$ in CRC cells. a** HCT116 cells were subjected to CRISPR/Cas 9 gene editing using RNP and ssODN to knock-in either the p53$^{R175H}$ or the p53$^{R273H}$ mutation. Cells which underwent the same process but did not end up with an edited genome, and thus retained wtp53 expression, served as CRISPR control. **b** RT-qPCR analysis of p21 mRNA in the cells in **a**. For the CRISPR/Cas9 knock-in clones, values in each experiment were determined separately for each individual clone, normalized to GAPDH mRNA, and then averaged. Values in the figure are displayed relative to the control parental cells, defined as 1.0. Mean ± SEM from four independent experiments. one-way ANOVA and Tukey's post hoc test. **c** RT-qPCR analysis of representative R273 signature genes in the cells in **a**. Values were calculated as in **b**. Three biological repeats (ITGA7,CDC42EP5), four biological repeats (APOE) or five biological repeats (MRC2, ECM1). **d** Western blot analysis of HT-29 cells transduced with p53-specific shRNA (Shp53) or control shRNA (Shc). $n = 2$. **e** Western blot analysis of COGA-5 cells transduced with p53-specific shRNA (Shp53) or control shRNA (Shc). $n = 2$. **f** RT-qPCR analysis of representative R273 signature genes in the cells in **d**. Values were normalized to GAPDH mRNA and are shown relative to the Shc cells. Mean ± SEM from three independent experiments. Unpaired two-tailed $t$ test. **g** RT-qPCR analysis of representative R273 signature genes in the cells in **e**. Values were normalized to GAPDH mRNA and are shown relative to the Shc cells. Mean ± SEM from four independent experiments. Unpaired two-tailed $t$ test. Source data is provided as a Source Data file.

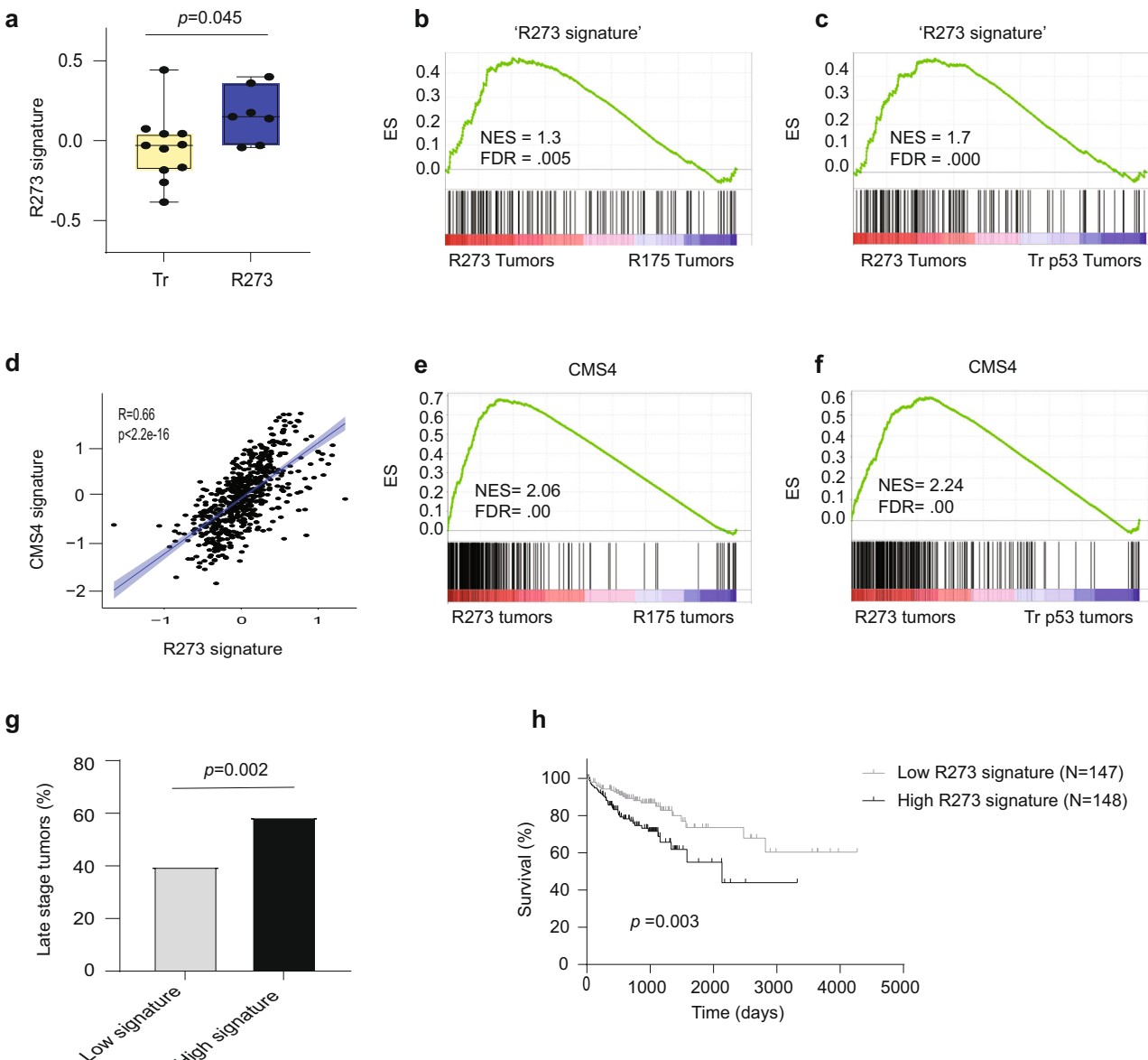

**Fig. 4 The R273 signature is upregulated in CRC cell lines and tumors and is associated with poor survival. a** Relative expression of the R273 signature in seven CRC cell lines harboring R273 mutations (SW480, SW620, CL14, HT-29, NCIH508, SNU503, SNUC2A) or truncating *TP53* mutations (Tr; n = 11). The boxplot displays data quartiles, horizontal lines mark the medians and upper and lower whiskers indicate maximum and minimum values for each distribution. Data accrued from Xena browser Cancer Cell Line Encyclopedia (CCLE) RNA-seq gene expression data (RPKM). Before mean expression calculation, all genes in the R273 signature were normalized to contribute equally to the signature. Unpaired two-tailed *t* test. **b**, **c** GSEA of CRC tumors harboring R273 mutations (n = 28) compared to tumors harboring R175 (n = 36) or truncating (Tr; n = 28) mutations; for truncating mutations, we selected the 28 samples with the lowest p53 mRNA levels, to better approximate null mutations. Genes were ranked by fold change, and the R273 signature was used as the tested gene set. **d** Pearson R correlation between the R273 signature and the cell-intrinsic gene signatures of the CMS4 subtype (Sveen et al., 2018). **e**, **f** GSEA of the same CRC tumors as in **b**–**c**, except that the CMS4 gene signature was used as the tested gene set. **g** Percentage of late-stage (stage 3–4) tumors among CRC tumors in the lowest quartile (N = 173) or highest quartile (N = 174) of R273 signature expression. **h** Overall survival of patients within the highest or lowest quartile of R273 signature expression in the TCGA colorectal cancer cohort. Log-rank test. Source data is provided as a Source Data file.

qPCR analysis of representative R273 signature genes confirmed that, in both cell lines, p53$^{R273H}$ selectively upregulated these genes, albeit to varying extents (Supplementary Fig. 3c, e). Moreover, using the cancer cell line encyclopedia (CCLE) database, we found that the R273 signature is significantly upregulated in CRC cell lines harboring R273 mutations, compared to CRC lines carrying protein-truncating *TP53* mutations (Fig. 4a). The CCLE includes only three R175-mutated CRC lines, precluding robust comparisons.

We next wished to extend these findings to human CRC tumors. Importantly, GSEA analysis of the TCGA CRC cohort revealed that tumors harboring R273 mutations displayed significantly higher expression of the R273 signature than those with R175 mutations (Fig. 4b and Supplementary Table 3). Comparison of the R273-mutated tumors to all tumors carrying truncating *TP53* mutations yielded a similar trend, but the difference did not reach statistical significance. However, the truncating mutations group is very heterogeneous, and not all

cases may resemble a true p53-null state. Yet, tumors with extremely low p53 mRNA levels, presumably owing to nonsense-mediated decay[3], are more likely to approximate true nulls. Indeed, when we included only truncating mutation cases displaying greatly reduced steady-state p53 mRNA, unequivocal association of R273-mutated tumors with the R273 signature was clearly evident (Fig. 4c and Supplementary Table 3). Interestingly, analysis of the entire set of CRC tumors revealed a remarkable degree of positive correlations between the expression levels of the genes comprising the R273 signature, which was not observed in three independent control signatures (Supplementary Fig. 4a, b). This suggests that many of the genes comprising the R273 signature may be subject to common transcriptional or post-transcriptional regulatory mechanisms.

Guinney et al. have recently employed comprehensive data analysis to define four consensus molecular subtypes (CMS) for colorectal cancer[35]. Remarkably, when we compared our R273 signature with the cell-intrinsic transcriptional signatures of the four CMS subtypes, as determined by Sveen et al.[36], the R273 signature displayed a strong (R = 0.66) and significant ($p < 2.2e-16$) correlation with the CMS4 signature (Fig. 4d). Furthermore, GSEA analysis confirmed that CRC tumors harboring R273 mutations are significantly associated with the CMS4 gene signature compared to tumors harboring R175 or truncating mutation (Fig. 4e, f). Interestingly, the GSEA analysis revealed that tumors harboring R175 mutations are significantly associated with the CMS2 gene signature, when compared to tumors harboring either R273 or truncating mutations (Supplementary Fig. 4c). Hence, R273 mutations and R175 mutations are differentially associated with distinct CRC molecular subtypes, possibly implicating them in different cancer-promoting biological processes[35].

Importantly, comparison of TCGA CRC tumors displaying high (upper quartile) expression of the R273 signature vs those with low (bottom quartile) expression revealed that high R273 signature expression was significantly associated with late-stage disease (Fig. 4g) and shorter patient survival (Fig. 4h). Furthermore, multivariate Cox regression analysis for overall survival, including age, sex, tumor location and the presence of KRAS mutations, demonstrated that high expression of the R273 signature is an independent prognostic factor (multivariate hazard ratio 2.314; 95% confidence interval 1.344–3.977; $P = 0.002$; Supplementary Table 4).

DNA contact mutation in arginine 248 of p53, particularly R248Q and R248W, are also very frequent in cancer (Supplementary Fig. 1a). To investigate whether those mutations endow p53 with the ability to regulate R273 signature genes, we stably expressed p53R248Q and p53R248W in p53KO SW480 cells (Supplementary Fig. 5a). As seen in Supplementary Fig. 5b, both mutants significantly upregulated representative R273 signature genes, to a similar extent as p53R273H. Moreover, CRC tumors harboring R248 mutations were significantly associated with enrichment of the R273 signature when compared to tumors harboring either R175 mutations or truncating mutations (Supplementary Fig. 5c–d). Concordantly, R248 mutations tend to be enriched in advanced CRC stages, albeit not as strongly as R273 mutations (Supplementary Fig. 5e), and are associated with reduced disease-specific survival than R175-mutated tumors (Supplementary Fig. 5f).

In sum, the R273 gene signature is broadly enriched in CRC cells and tumors harboring the most common DNA contact mutations, and is correlated with shorter patient survival. This further supports the hypothesis that the transcriptional output directed by such mutants endows CRC tumors with more aggressive features, which adversely affect patient outcome.

**R273 mutants selectively promote cell spreading, migration and invasion.** To elucidate oncogenic pathways that may contribute to the clinical impact of R273 mutations, we subjected the R273 signature to Gene Ontology analysis by METASCAPE[37]. Interestingly, many observed pathways were directly or indirectly related to cytoskeleton dynamics (Fig. 5a), which is often associated with cancer-related properties such as cell adhesion, spreading, migration and invasion[38–41]. Specifically, the Rho signaling pathway, ranking high in this analysis, can promote cancer by driving actin cytoskeleton remodeling and augmenting cell migration, survival, polarity, and more[42,43].

Phenotypically, the morphology of SW480 cells expressing p53R273H differed visibly from that of parental knockout cells or p53R175H overexpressors. This was evident as accelerated spreading, confirmed by time-lapse microscopy (Fig. 5b and Supplementary Movies 1–3). Similar observations were made with RKO cells, depleted of their endogenous wtp53 and reconstituted with either p53R175H or p53R273H (Supplementary Fig. 6a). Moreover, RNA-seq analysis six hours after plating (Supplementary Fig. 6b) showed that already at this early time point the R273 signature was upregulated in the p53R273H expressors to a similar extent as after 24 h, supporting the notion that the inherent gene expression pattern dictated by p53R273H drives cell spreading, rather than being a consequence of spreading.

Cell cycle analysis did not reveal differences between the effects of p53R273H and p53R175H when overexpressed in SW480 cells (Supplementary Fig. 6c). However, the p53R273H overexpressors displayed a significant increase in cell migration, relative to either p53R175H overexpressors or p53KO cells (Fig. 5c, d). Importantly, parental SW480 cells (expressing p53R273H and p53P309P-) also migrated faster than the p53KO cells (Supplementary Fig. 6d, e). Likewise, the HCT116 p53R273H knock-in clones migrated significantly faster than either control wtp53-expressing HCT116 cells or the p53R175H knock-in clones (Fig. 5e, f). The p53R273H overexpressing SW480 cells were also more invasive than the p53KO and p53R175H overexpressing cells (Fig. 5g). Moreover, while both p53R273H and p53R175H augmented the migration of p53-depleted RKO cells and p53-truncated COLO-205 cells (Supplementary Fig. 6f–i) and the invasiveness of HCT116 knock-in cells (Fig. 5g), the effect of p53R273H was greater. Thus, p53R273H preferentially promotes cell spreading, migration and invasion.

Rho signaling is one of the top enriched pathways in the R273 signature (Fig. 5a). In agreement, a Rho proteins GTPase activation assay confirmed that p53R273H overexpression in SW480 cells augmented the activation of both Cdc42 and Rac1, relative to p53R175H overexpressors (Fig. 5h). Interestingly, RhoA activation was not differentially affected. Importantly, the migratory phenotype of p53R273H overexpressors was completely abolished by treatment with the Rac1/Cdc42 inhibitor MBQ-167 (Fig. 5i). Hence, p53R273H selectively drives Rac1/Cdc42-dependent cancer cell migration.

**p53R273H preferentially promotes metastasis.** We next wished to assess whether the differential impact of p53R273H in vitro is also reflected in a more aggressive phenotype in vivo. To that end, SW480 cells overexpressing either p53R175H or p53R273H were injected into the tail vein of NSG mice (Fig. 6a). Remarkably, 9 weeks after injection, mice injected with p53R273H-overexpressing cells displayed a significantly larger total area of lung metastases than mice injected with p53R175H overexpressors (Fig. 6b, c). Moreover, to better recapitulate CRC biology, we orthotopically injected SW480 cells harboring the two p53 mutants into the cecal wall of NSG mice (Fig. 6d). Seven weeks later, mice were sacrificed

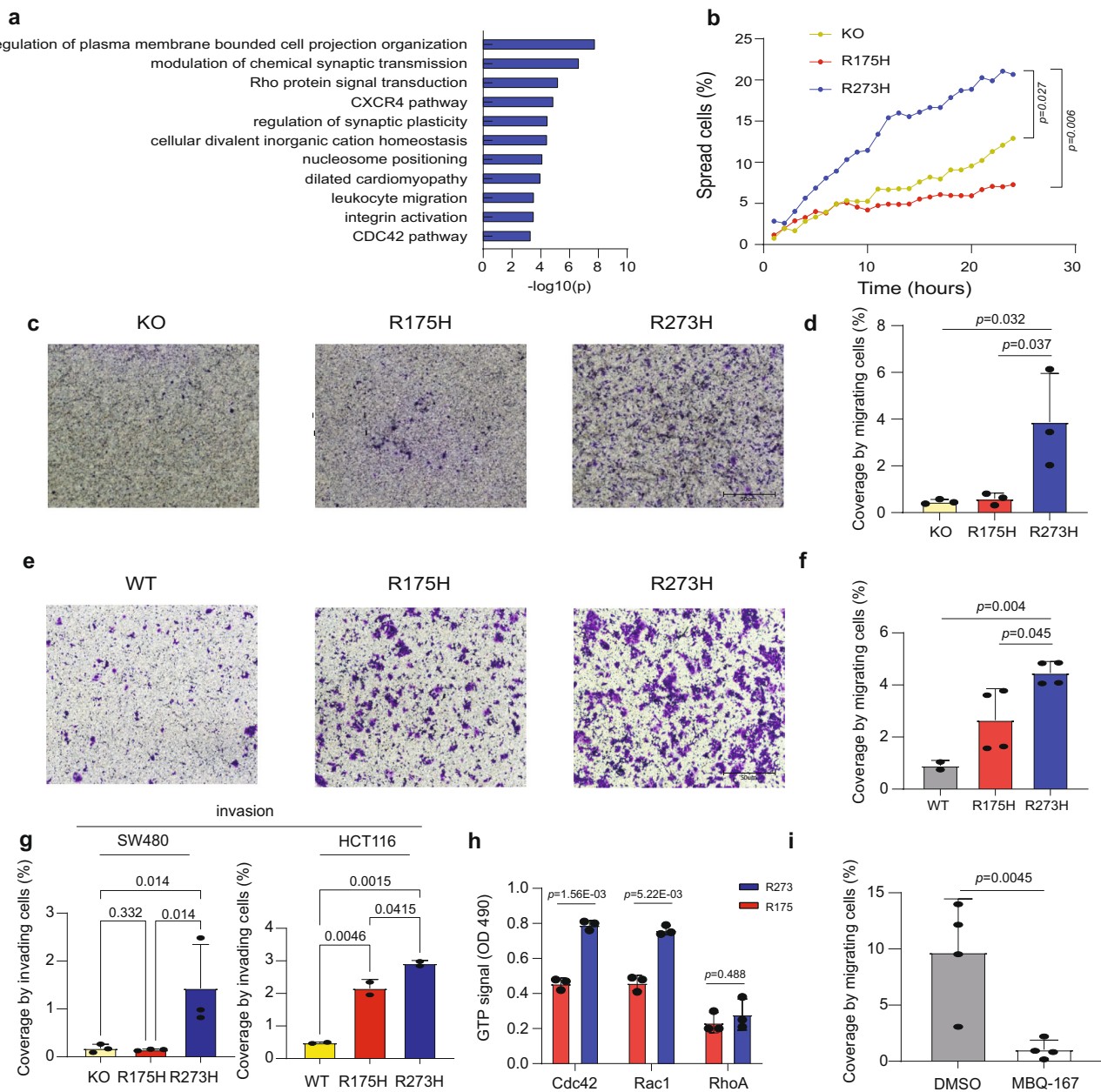

**Fig. 5 p53^R273H promotes cell spreading, migration and invasion. a** Gene Ontology analysis of the R273 signature (Metascape). **b** Kinetics of spreading of SW480 p53 KO cells (KO) and their derivatives stably overexpressing p53$^{R175H}$ or p53$^{R273H}$. Percentages of spread cells in the course of 24 h were determined by time-lapse microscopy. Images were taken at 1 h intervals, and were subjected to cell segmentation and aspect ratio calculation. Statistical analysis at t = 24 was done using one-way ANOVA and Tukey's post hoc test. Two biological repeats. **c** Representative images of transwell migration assays performed with SW480 p53 KO cells and their derivatives stably overexpressing p53$^{R175H}$ or p53$^{R273H}$, taken 24 h post-seeding. **d** Average percentage of coverage (ImageJ) by migrating cells in transwell migration assays as described in **c**. Mean ± SEM from Three biological repeats. Nested one way ANOVA and Tukey's post hoc test. **e** Representative images of transwell migration assays performed with HCT116 CRISPR/Cas9 control cells (WT) or CRISPR/Cas9 knock-in of either p53$^{R175H}$ or p53$^{R273H}$. An equal number of cells from each of the 5 clones harboring the same mutation were pooled together and grown for one week prior to the migration assay. **f** Average percentage of coverage (ImageJ) by migrating cells in transwell migration assays as described in **e**. Mean ± SEM from four biological repeats. Nested one way ANOVA and Tukey's post hoc test. **g** Average percentage of coverage (ImageJ) by invading cells in transwell Matrigel invasion assays performed with the same cells as in **c** and **e**. Mean ± SEM from three biological repeats (SW480) or two biological repeats (HCT116). Nested one way ANOVA and adjustment for multiple comparison. **h** SW480 cells stably overexpressing p53$^{R175H}$ or p53$^{R273H}$ were subjected to Rho signaling activation analysis using a G-LISA assay kit. Mean ± SEM from Three technical repeats. **i** SW480 p53 KO cells stably overexpressing p53$^{R273H}$ were treated for 4 h with either DMSO or MBQ-167 (750 nM), and then subjected to a transwell migration assay as in **c**. Average percentage of coverage by migrating cells (ImageJ) is shown. n = 4. Nested one way ANOVA and Tukey's post hoc test. Source data is provided as a Source Data file.

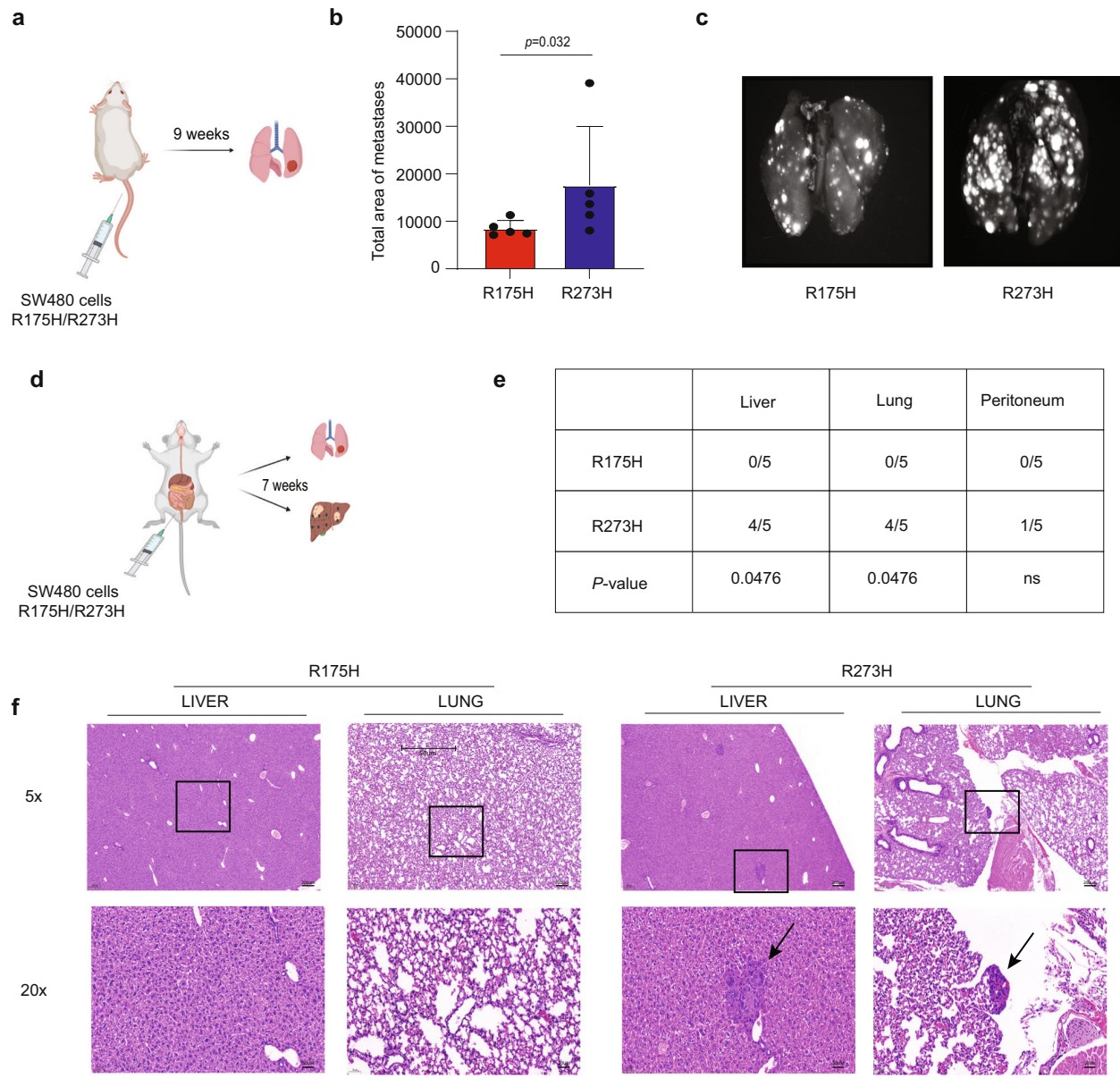

**Fig. 6 p53^R273H preferentially promotes metastasis. a** SW480 p53 KO cells stably overexpressing p53^R175H or p53^R273H were GFP labeled and injected into the tail vein of NSG mice. Lung metastases were visualized nine weeks post-injection. **b** Total area of metastases at the lung surface (calibrated units), as quantified with ImageJ (Mean ± SEM, 5 mice per group). Two-tailed Mann–Whitney *U*-test. **c** Representative images of lung metastases in mice analyzed as in **a**. **d** SW480 p53 KO cells stably overexpressing p53^R175H or p53^R273H were injected into the cecal wall of NSG mice. 7 weeks post-injection, Metastases were evaluated by a pathologist, using H&E-stained histology slides. **e** Numbers of mice with liver, lung, and peritoneal metastases in the groups described in **d**. **f** Representative H&E staining images of lung and liver tissue of mice analyzed as in (**d**). The bottom row shows a 20X magnification of the areas marked by squares in the 5X magnification images in the upper row. Arrows indicate metastatic foci. Source data is provided as a Source Data file.

and evaluated for distant organ metastases. Notably, four out of five mice in the R273H group developed both lung and liver metastases, while no metastases were observed in any of the mice injected with p53^R175H overexpressors (Fig. 6e, f). Thus, p53^R273H preferentially promotes metastatic behavior in vivo.

**p53^R273H is recruited to R273 signature genes and activates them via its transactivation domain.** To explore the molecular mechanisms driving the transcriptional upregulation of R273 signature genes by p53^R273H, we interrogated published p53 CHIP-seq data of SW480 cells[33], expressing endogenous p53^R273H (along with p53^P309S). Remarkably, analysis of all mutp53 peaks

using GREAT[44] revealed that the most significantly enriched cellular components associated with those peaks were related to cytoskeleton structure and function (Fig. 7a). Moreover, the mutp53 chromatin binding peaks were significantly positively correlated with the p53^R273H-upregulated genes in our RNA-seq (Fig. 7b), suggesting that upregulation of gene expression by p53^R273H is mediated, at least in part, via selective recruitment of p53^R273H to the corresponding chromatin regions. To query experimentally this notion, we compared by ChIP-qPCR the binding of p53^R273H and p53^R175H to regulatory elements of representative R273 signature genes in SW480 cells overexpressing either mutant. As seen in Fig. 7c, p53^R273H indeed displayed significantly stronger binding than p53^R175H to those regulatory regions.

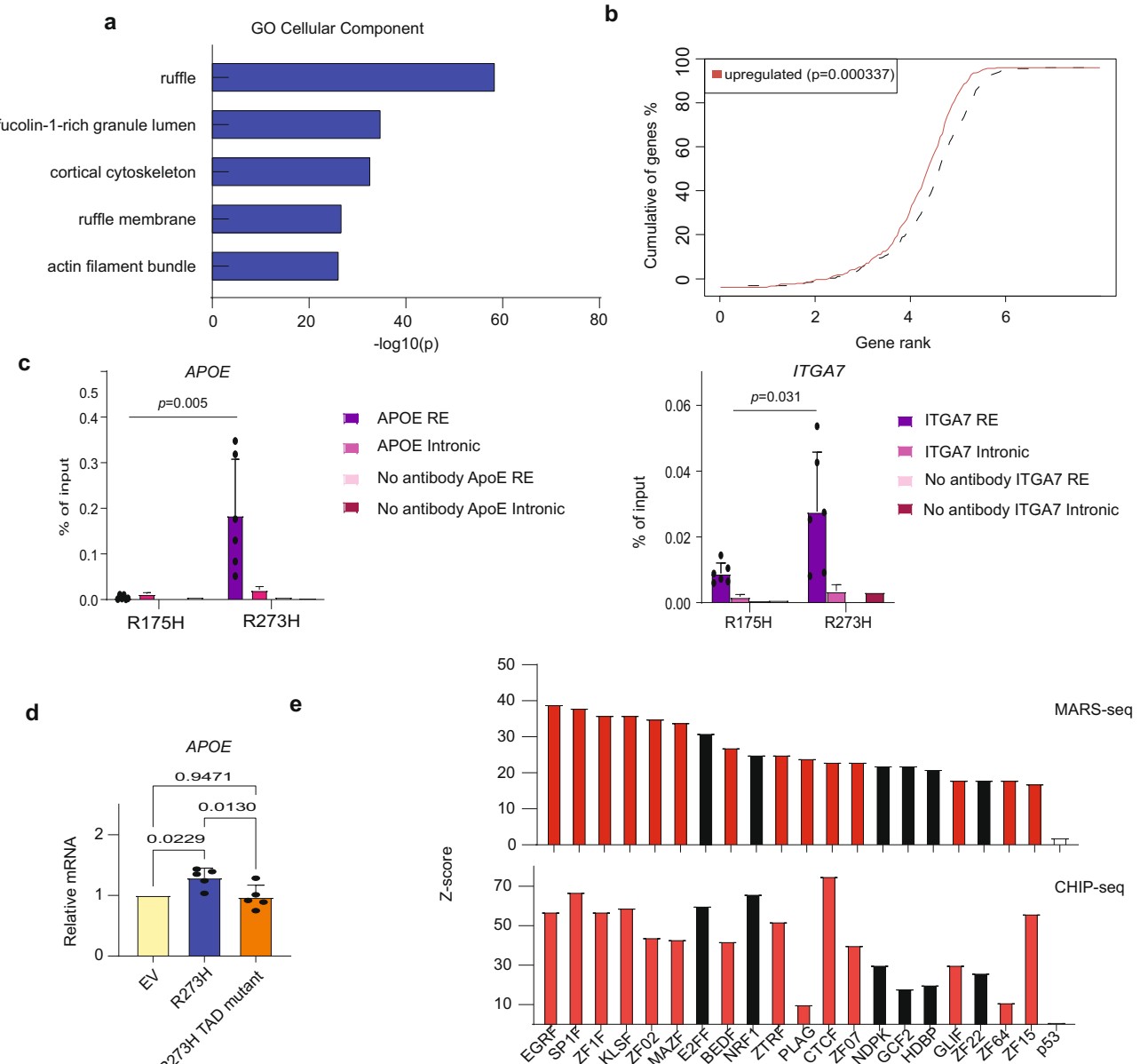

**Fig. 7 p53$^{R273H}$ binds gene regulatory elements and augments transcription. a** Top five enriched GO cellular components associated with endogenous mutp53 ChIP-seq peaks in SW480 cells. Data from Rahnamoun et al.[33], was subjected to analysis by GREAT as described in Methods. **b** Mutp53 chromatin binding peaks in SW480 cells are significantly associated with genes upregulated by p53$^{R273H}$. All individual genes were ranked by their distance to the nearest p53 ChIP-seq peak in Rahnamoun et al.[33]; the X-axis represents log 10 of the rank. Red line represents the genes upregulated in SW480 *TP53* KO cells stably transduced with p53$^{R273H}$, relative to control KO cells and cells transduced with p53$^{R175H}$ (see Fig. 2d). Dashed line indicates all the other, non-differentially expressed genes as background. One tailed Kolmogorov-Smirnov test. **c** ChIP-qPCR analysis of mutp53 binding to regulatory regions of representative R273 signature genes in SW480 cells stably overexpressing either p53$^{R175H}$ or p53$^{R273H}$. Binding of mutp53 to regulatory elements of *ITGA7* and *APOE* is compared to binding to intronic regions of the same genes. Nested one way ANOVA and Tukey's post hoc test. Three biological repeats. Total 6 repeats. **d** RT-qPCR analysis of *APOE* mRNA in SW480 *TP53* KO cells transiently transfected with empty vector control (EV), intact p53$^{R273H}$, or p53$^{R273H}$ harboring two mutations (L22Q and W23S) within the p53 transactivation domain (R273H TAD mutant). Values were normalized to GAPDH mRNA and are shown relative to the empty vector control cells. Mean ± SEM from five independent biological repeats (one-way ANOVA and Tukey's post hoc test). **e** Transcription factors (TF) binding sites overrepresented in canonical promotors of the R273 signature genes. Upper panel shows the top 20 TFs enriched in the R273 signature gene promoters relative to all canonical gene promotors. Lower panel shows the extent of overrepresentation of the same 20 TFs, at mutp53 binding sites in SW480 cells, determined experimentally by Rahnamoun et al.[33], relative to the entire human genome sequence. wtp53 is included in both panels as an example of a non-enriched TF. "F" in EGRF, SP1F etc. relates to "family". Red bars indicate zinc finger transcription factors. Source data is provided as a Source Data file.

Previous work has demonstrated that p53$^{R273H}$ can act as a potent transcriptional activator when recruited to DNA[45–48]. The N-terminal transactivation domain (TAD) is essential for this activity[47,48]. In agreement, while transiently-transfected p53$^{R273H}$ augmented the expression of endogenous R273 signature genes in p53KO SW480 cells, a TAD-mutated version of p53$^{R273H}$, despite being expressed at comparable amounts in the transfected cells (Supplementary Fig. 7a, b), was incapable of such transcriptional augmentation (Fig. 7d, Supplementary Fig. 7c). The p53$^{R273C}$ mutation is also fairly common in human cancer, including CRC.

As seen in Supplementary Fig. 7a–c, similarly to p53[R273H], p53[R273C] also transactivated endogenous R273 signature genes in transiently transfected p53KO SW480 cells. Concordantly, the two R273 mutants are also associated with similar disease-specific survival of CRC patients (Supplementary Fig. 7d).

As mutp53 is unlikely to bind directly to DNA[5,7], its recruitment to those regions is probably mediated by other transcription factors (TFs). Computational analysis of the DNA sequences of the R273 signature gene promoters suggested that they are enriched for putative binding sites of numerous TFs (Fig. 7e, upper panel). Reassuringly, binding sites of the majority of those TFs are also predicted to be enriched in regions comprising the mutp53 binding peaks mapped experimentally by Rahnamoun et al.[33] (Fig. 7e, lower panel). This suggests that at least some, if not most, of those TFs may serve as anchors for recruitment of p53[R273H] to the promoters of R273 signature genes. Interestingly, many of those TFs bind specifically to GC-rich DNA sequences[49] and contain CpG dinucleotides within their recognition motif[50]. Congruently, the promoters of the R273 signature genes were found to be highly enriched for CpG islands (Supplementary Fig. 7e). It is conceivable that the ability of R273 mutants to bind these regions may be modulated by specific epigenetic changes, which might lead to context-dependent upregulation of R273 signature genes. Collectively, these observations support the notion that recruitment of R273-mutated p53 proteins to specific chromatin regions alters the expression of associated genes, in a TAD-dependent manner. These transcriptional alterations may underpin the observed biological effects of the R273 mutants, leading to enhanced tumor progression and worse patient outcome.

## Disscusion

The abundance of *TP53* mutations and the increasing amount of clinical and genomic data derived from cancer patient tumors represent an opportunity to better understand the impact of different *TP53* mutants on the features of the tumors that harbor them. Such understanding may potentially help in translating *TP53* status information into better individualized treatment decisions. This is particularly relevant for CRC, where the frequency of *TP53* missense mutations, and especially hotspot mutations, is very remarkable.

In the present study, we compared the effects in CRC of two prevalent *TP53* mutations, representing distinct types of mutp53 proteins. We show that R273 mutations direct a unique transcriptional program, which is not expressed in p53-null CRC cells or in tumors harboring truncating *TP53* mutations, and thus constitutes a GOF activity of R273 mutants. Importantly, this program, which entails activation of critical cancer-related pathways associated with cytoskeleton function, cell invasion and metastatic properties, while being enriched also in CRC tumors harboring DNA contact mutations at position R248 of p53, is not shared with R175 mutants. This corresponds to clinical data from multiple CRC cohorts, suggesting that R273 and R248 mutants are associated with accelerated cancer progression and overall more aggressive disease. Mechanistically, induction of this transcriptional program by R273 mutants appears to entail their differential recruitment to specific regulatory elements on the DNA. Most probably, such recruitment is not direct, relying on the preferential association of R273-mutated p53 with sequence-specific DNA binding proteins[7,51–53].

Although many of the published studies on mutp53 GOF have focused on common features shared by multiple mutants[54–59], differential effects of different hotspot mutants have also been described[53,60–62], including quantitative differences in their interaction with critical partner proteins[63,64]. Of note, a recent study employing HCT116 CRC cells showed that p53[R273H] is a more potent enhancer of cancer cell stemness than other p53 hotspot mutants, owing to selective regulation of a subset of long noncoding RNAs[65]. We now show that the differences between mutants go beyond molecular features and may actually dictate different patient survival. Moreover, we show that selective mutp53 GOF effects can be abolished by a specific pathway inhibitor, suggesting that patients whose tumors harbor different p53 mutants might react differently to the same treatment protocol. Hence the particular *TP53* mutation, not just the presence or absence of *TP53* mutations, may be of future value when devising individualized treatment strategies for CRC, and most probably also for other cancer types.

Surprisingly, in our study p53[R175H] did not exert measurable effects on the transcriptional landscape and biological features of SW480 cells. This was unexpected, given that R175 mutations are very frequent in CRC: if they have no contribution to this type of cancer, why are they seen so often? A trivial explanation might be that they merely occur at high frequency because of particular mutation signatures inherent to CRC, without any acquired GOF[10]. Yet, a more appealing possibility is offered by the fact that R175 mutations are strongly associated with the CMS2 transcriptional signature (Fig. S5c). CMS2 tumors are characterized by WNT and MYC signaling activation[35]. If p53 R175 mutants facilitate such activation, they are expected to promote CRC initiation and rapid primary tumor growth. Indeed, R175 mutations are more prevalent than R273 mutations in early stages of the disease, but become less prevalent at late stages, when invasive and metastatic capacities take the lead role (Fig. 1b). Furthermore, CMS2 tumors tend to be more "immune cold", displaying minimal expression of immune-related transcripts and low infiltration of immune cells[66,67]. It is conceivable that this may be partly due to GOF effects of mutp53, as suggested recently for pancreatic cancer[57]. In such scenario, one might propose that R175 mutants may be particularly potent facilitators of immune evasion at early stages of CRC development, favoring their high abundance at those stages.

Still, it is surprising p53[R175H] hardly affected the SW480 transcriptome, despite being abundantly expressed. The most plausible explanation is that the effects of distinct p53 mutants are highly context-dependent. SW480 cells possess endogenous p53[R273H] (as well as p53[P309S]) and their transcriptional profile is consistent with the R273 signature and hence with the CMS4 program. Presumably, their intrinsic signaling context has been evolutionarily optimized to support the transcriptional and biological GOF effects of their endogenous p53[R273H], while concomitantly becoming non-supportive of alternative programs driven by other mutants such as p53[R175H], which are characteristic of CMS2 tumors. This conjecture is in line with broader evidence for context-dependent GOF effects of missense mutp53 proteins. For example, whereas a particular subset of p53 mutants is selectively enriched experimentally in vivo, consistent with GOF, these mutants are not enriched and do not reveal any GOF properties when the same cells are grown in vitro[68,69]. A striking example of the context dependency of p53 mutations in CRC has recently been described by showing that the gut microbiome can dictate whether mutp53 proteins enhance tumor growth or, conversely, even restrict it, displaying surprising tumor suppressor features[70]. Intriguingly, even the R273 mutant, which we show here to exert distinct GOF effects, did not exhibit measurable GOF effects in a genetically modified mouse model of CRC[71], further demonstrating that the contribution of a particular p53 mutation to cancer progression is highly context-dependent.

The benefit of adjuvant therapy for colon cancer patients with stage 2 tumors remains unclear. Decisions regarding adjuvant

therapy presently involve assessment of recurrence risk, based on clinicopathological features[72]. Our data suggests that CRC patients with R273 mutation are more prone to advance to late stage disease and therfore are more likely to benefit from early adjuvant therapy. Given that *TP53* mutations are the most frequent single gene mutations in human cancer and that practically all current analytical cancer gene panels include *TP53*, this provides an opportunity to improve treatment decisions for stage 2 colorectal patients.

Altogether, our findings argue that different p53 mutants may impart non-identical features on tumors, eventually impacting patient outcome. Better understanding of such differential contributions of distinct p53 mutants and their context dependency is bound to make information on *TP53* mutations more valuable and may enable better precision-based medicine in the future.

## Methods

**Data acquisition and processing**. *TP53* somatic mutation status and clinical attributes from the DFCI, CPTAC-2 and MSKCC cohorts[22,23,25] were retrieved from the CBioPortal open Platform. *TP53* somatic mutation status and clinical attributes from the TCGA and ICGC (CRC cohorts) were downloaded from UCSC Xena Browser http://xena.ucsc.edu/. *TP53* somatic mutation status and clinical attributes from GECCO and CCFR were taken from published data[24]. All patients were grouped according to their *TP53* status.

TCGA RNA-Seq expression profiles ((HT-Seq count, log2(fpkm-uq+1) for normalization)), were downloaded from UCSC Xena Browser http://xena.ucsc.edu/. TCGA colon adenocarcinoma (TCGA-COAD) and rectal adenocarcinoma (TCGA-READ) samples were filtered for primary tumour samples and divided according to their *TP53* status. Truncating mutation tumors were defined as tumors with *TP53* frameshift, nonsense and splice site mutations.

RNA-seq data and gene somatic mutations data from cancer cell lines was downloaded from Xena Browser (CCLE dataset, RPKM), filtered for large intestine cell lines and divided into groups according to their *TP53* status.

**Cell lines, transfections and viral infections**. Cells were maintained at 37 °C with 5% $CO_2$. SW480 and RKO cells were cultured in DMEM (Biological Industries, BI), COLO-205 cells were grown in RPMI (BI) and HCT116 cells were grown in McCoy's 5 A (Sigma). All culture media were supplemented with 10% FBS (BI) and 1% penicillin–streptomycin (BI). All cell lines tested negative for Mycoplasma. SW480 *TP53* knockout cells and RKO *TP53* knockout cells, generated as described previously[31,34], were a kind gift from Varda Rotter (Weizmann Institute of Science).

Plasmid transfection was done with the jetPEI DNA transfection reagent (Polyplus Transfection). The final DNA amount was 2 µg per well in a 6-well dish, and the transfection medium was replaced after 24 h. Cells were collected 48 h after transfection for gene expression profiling by RT-qPCR. pCB6, pCB6-R273H, pCB6-R273C and pCB6-R273H with substitutions of residues 22 and 23 (L22Q/W23S; R273H TAD mutant), were a generous gift from Karen Vousden.

For stable gene transduction, SW480 p53KO cells, RKO p53KO cells and CACO-205 cells were infected with recombinant lentiviruses (pEF1alpha-p53 R273H IRES-EGFP and pEF1alpha-p53R175H IRES-EGFP), to express the corresponding mutant p53 proteins. Lentiviral packaging was performed by jetPEI-mediated transfection of Phoenix cells with the indicated plasmid DNAs, together with a plasmid encoding the VSVG envelope protein and packaging plasmids. Virus-containing supernatants were collected 48 h and 72 h after transfection, filtered, and supplemented with 8 µg/ml polybrene (Sigma). One week post infection, cells were subjected to FACS sorting for GFP positive cells. Alternatively, SW480 cells were infected with recombinant lentiviruses (pLKO.1-puro-shp53, TRCN0000010814 (Sigma)) to produce shRNA directed against the 3' UTR of the endogenous mutant p53 mRNA, together with recombinant lentiviruses (pEF1alpha-p53 R273H IRES-EGFP and pEF1alpha-p53R175H IRES-EGFP) to express the corresponding mutant p53 proteins. 48 h after infection, p53 knockdown cells were selected with puromycin, and one week later were subjected to FACS sorting for GFP positive cells. HT-29 and COGA-5 were infected with recombinant lentiviruses (pLKO.1-puro-shp53, (addgene, 19199)) to produce shRNA directed against the endogenous mutant p53 mRNA. p53 knockdown and mutant protein expression were verified by RT-qPCR and Western blot analysis.

**CRISPR/Cas9-mediated homology-directed repair (HDR)**. HCT116 cells (ATCC) were edited by CRISPR-HDR as previously described[73], with some modifications. First, an RNP complex was prepared by mixing recombinant Alt-R® *Streptococcus pyogenes* Cas9 V3 protein (104 pmol, IDT) with Alt-R® single guide RNA (260 pmol, IDT). After 15 min at RT to allow the formation of the complex, the RNP was added to 200,000 HCT116 cells which had been harvested before, washed and resuspended in 20 microliter of SE Cell Line Nucleofector® Solution (Lonza). Next, 120 pmol of the Alt-R® HDR single-stranded oligodeoxynucleotide

(IDT) was added and cells were transferred to an Amaxa 4D Nucleofector (Lonza). Electroporation was carried out using cell line-specific settings according to the manufacturer's recommendations (EN-113). Cells were then transferred to a recovery plate with fresh medium and HDR enhancer compound Alt-R™ HDR Enhancer V2 (1.0 µM, IDT). After a few days of recovery, cells were seeded as single cells in 96 well dishes and genome-edited clones were identified by Sanger sequencing. The sequences of the guide and repair oligonucleotides are listed in Supplemental Table 5.

**Immunoblotting**. Cell pellets were resuspended in RIPA buffer, and protein sample buffer was added after centrifugation. Samples were boiled and resolved by SDS-PAGE. The following antibodies were used: GAPDH (Cell Signaling, 14C10, 1:1000), p53 (mixture of monoclonal antibodies DO1 + PAb1801). Imaging and quantification were performed using a ChemiDoc MP Imager with Image Lab 4.1 software (Bio-Rad).

**Time-lapse microscopy**. Cells were plated in 6 well plastic bottom dishes and monitored by time-lapse imaging using a Celldiscoverer 7 microscope (Carl Zeiss Ltd.) Imaging was performed using the oblique contrast method through a Plan-Apochromat 20X/0.7 and a 0.5x Tubelens (effective magnification of 5X and 0.35NA). Illumination was done with a white-light LED set to 10% and detection was by a 14 bit Axiocam 506 CCD camera (Carl Zeiss Ltd.) with 10 ms exposure time. Pixel size was 0.462 m × 0.462 m. Image tiling was used in order to cover a large area. Images were taken at 1 h intervals, for a total of 24 h.

To quantify the cell shape, we segmented the cells using the ilastik Boundary based segmentation with Multicut workflow[74]. We trained in ilastik (1) auto-context pixel classifier for 3 classes: boundary/cell/background and (2) multi-cut edge classifier. These were then applied sequentially to all the images in batch. We wrote a Fiji[75] macro to select cells from the multi-cut objects based on their size (between minimum and maximum values) and their average probability of belonging to the "cell" class of the ilastik auto-context pixel classifier. We discarded cells touching the border of the image. For each cell, we measured the aspect ratio (AR) – the ratio between the major and minor axis of the best-fitted ellipse. Spread cells were defined as those with AR > 1.8. For each time point, the percentage of spread cells out of the total number of detected cells was calculated.

**MARS-seq**. SW480 p53KO cells and their derivatives stably overexpressing p53$^{R273H}$ and p53$^{R175H}$ were seeded at a density of 1.5 million per 10 centimeter dish. RNA was extracted either 6 h or 24 h post seeding, using a NucleoSpin kit (Macherey Nagel). RNA of SW480 cells with stable p53 knockdown or over-expression of shRNA-resistant p53$^{R175H}$ or p53$^{R273H}$ was extracted similarly.

MARS-seq libraries were prepared at the Crown Genomics Institute of the Nancy and Stephen Grand Israel National Center for Personalized Medicine, Weizmann Institute of Science. A bulk adaptation of the MARS-Seq protocol[32] was used to generate RNA-seq libraries for expression profiling. Briefly, 30 ng of input RNA from each sample was barcoded during reverse transcription and pooled. Following Agencourt Ampure XP beads cleanup (Beckman Coulter), the pooled samples underwent second strand synthesis and were linearly amplified by T7 polymerase in vitro transcription. The resulting RNA was fragmented and converted into a sequencing-ready library by tagging the samples with Illumina sequences during ligation, RT and PCR. Libraries were quantified by Qubit and TapeStation as well as by qPCR for GAPDH as previously described[32]. Sequencing was done with a Nextseq 75 cycles high output kit (Illumina). Differential expression was analyzed using the UTAP pipeline[76].

Heatmaps were generated with Partek Genomics Suite 7.0 (Partek Inc.), using log normalized values (rld), with row standardization and Euclidean clustering.

**Gene set enrichment analysis**. Gene Set Enrichment Analysis (GSEA)[77], was employed to determine whether the R273 gene signature exhibits a statistically significant bias in its distribution within a ranked gene list. We followed the standard procedure as described in the GSEA user guide (http://www.broadinstitute.org/gsea/doc/GSEAUserGuideFrame.html) to create the ranked gene list for RNA-seq profiling of our data/published data/TCGA data, and tested the R273 signature for significant differences in distribution. The FDR for GSEA is the estimated probability that a gene set with a given NES (normalized enrichment score) represents a false-positive finding.

**RT-qPCR**. RNA was isolated using the NucleoSpin kit (Macherey Nagel). 1 µg of each RNA sample was reverse transcribed using Luna® Universal qPCR Master Mix (New England Biolabs). Real-time qPCR was performed using SYBR Green PCR Supermix (Invitrogen) with a StepOne real-time PCR instrument (Applied Biosystems). For each gene, values for the standard curve were measured and the relative quantity was normalized to *GAPDH* mRNA. Primers are listed in Supplementary Table 6.

**RhoGTPase activity assay**. Endogenous RhoA, Rac1 and Cdc42 activity levels were determined by using an enzyme-linked immunosorbent assay (ELISA)-based G-LISA kit (Cytoskeleton, Inc #BK135) strictly following the manufacturer's

instructions. Briefly, SW480 cells stably overexpressing p53[R175H] or p53[R273H] were plated and allowed to grow to ~70% confluence before being washed with PBS and lysed in 100 μl of ice-cold lysis buffer in the presence of protease and phosphatase inhibitors. The lysate was clarified by centrifugation at $10,000 \times g$ for 1 min, and snap-frozen in liquid nitrogen. After normalizing protein concentration using PrecisionRed (Cytoskeleton, Inc), samples were added in triplicate to wells coated with a respective GTP-binding protein. After washing, bound GTPases levels were determined by subsequent incubations with a respective antibody and a secondary HRP-conjugated antibody, followed by addition to an HRP detection reagent. Background was determined by a negative control well. Absorbance was measured at a wavelength of 490 nm using a microplate reader (Thermo Fisher Scientific). Values are expressed as mean ± SEM of three technical replicates.

**Migration assays**. Migration assays were performed using the transwell system (8 μm pore size; Costar). In brief, 60,000 cells in either serum-free medium (RKO), medium containing 1% FBS (SW480) or 2% FBS (COLO-205 and HCT116) were seeded in the upper chamber, while the lower chamber was filled with 600 microliter of culture medium supplemented with either 10% FBS (RKO, SW480), or 2% FBS supplemented with 10 ng/ml EGF as chemoattractant (COLO-205, HCT116). Cells were allowed to migrate for 24 h (SW480, COLO-205, and HCT116) or 30 h (RKO). Cells on the lower surface of the chamber were fixed with 4% PFA and stained with crystal violet. Cells on the upper surface were removed with cotton plugs. Stained cells were imaged with a Nikon Eclipse Ti-E microscope at ×4 magnification, capturing at least three fields for each condition, and crystal violet stained areas were quantified with an ImageJ macro. Coverage by migrating cells was calculated as percentage of stained area relative to total area.

For MBQ-167 migration assay, SW480 cells were treated for 4 h with either MBQ (750 nM) or DMSO. After 4 h, cells were trypsinized and placed in the upper chamber as above. 600 microliter of culture medium containing 10% FBS and either MBQ-167 (750 nM) or DMSO was added to the bottom chamber. 24 h post seeding, cells were fixed and stained. The stained area was quantified as above.

**Invasion assays**. For invasion assays, 200,000 cells were seeded in transwell chambers pre-coated with Matrigel (Corning). 600 microliter of culture medium containing 10% FBS and supplemented with EGF (10 ng/ml) was added to the bottom chamber. After 24 h, cells were fixed and stained. The stained area was quantified as above.

**In vivo experiments**. All animal experiments and methods were approved by the Weizmann Institutional Animal Care and Use Committee (approval 07200820-3). The Weizmann Institutional Animal Care and Use Committee does not permit experiments were tumors reach 10% of normal body weight and this size was not exceeded in our experiments. For tail vein injection, $2.5 \times 10^6$ cells were resuspended in 100 microliter PBS before being injected through the tail vein to 10, 8 weeks C.B-17/IcrHsdPrkdc-scid-Lyst-bg female mice. Tumors were harvested 9 weeks post-injection. For orthotopic injection, $1 \times 10^7$ cells were re-suspended in 50 microliter PBS, diluted in Matrigel (1:1), and injected into the cecal wall of 10, 8 weeks C.B-17/IcrHsdPrkdc-scid-Lyst-bg female mice. Tumors were harvested 7 weeks post-injection.

**Chromatin immunoprecipitation (ChIP) analysis**. Chromatin immunoprecipitation was performed as previously described[51]. SW480-p53[R175H] and SW480-p53[R273H] cells at 70% confluence were subjected to crosslinking by adding 1/10 volume of fresh 11% formaldehyde solution (50 mM HEPES-KOH pH7.5, 100 mM NaCl, 1 mM EDTA, 0.5 mM EGTA, 11% formaldehyde) for 10 min, followed by incubation in 0.125 M glycine for 5 min. DNA was sheared to a range of 100–600 bp by subjecting the chromatin to sonication in a Bioruptor sonicator (Diagenode). 1/10 of the chromatin sample was set aside as input. Mouse anti-p53 antibody (Santa Cruz, DO1, sc-126) and normal mouse IgG (Santa Cruz, sc-2025) were used for immunoprecipitation. Immune complexes were collected using Dynabeads protein G (Thermo Fisher Scientific). After reverse crosslinking and Proteinase K digestion, DNA was recovered using ChIP DNA Clean & Concentrator columns (Zymo Research). qPCR was performed using Luna® Universal qPCR Master Mix (New England Biolabs) on a 7500 Fast Real-Time PCR System (Thermo Fisher Scientific). Data was normalized by the ΔΔCt method over Input (1:20 dilution) and IgG samples. Sequences of the primers used for ChIP analysis are listed in Supplementary Table 6.

For Genomic Regions Enrichment of Annotations Tool (GREAT), we used published ChIP-seq data (GEO Series Accession Number GSE102796). Fastq files were downloaded from GEO and analyzed using the UTAP pipeline[76]. 17,980 peaks identified in two replicates were analyzed for GO cellular component enrichment using GREAT[44]. The ChIP-seq peaks were integrated with differential gene expression from MARS-seq using the BETA tool (http://cistrome.org/ap/root)[78]. BETA basic was used to perform factor function prediction (up and down-regulation) and direct target detecting, with a distance of up to 10,000 bp between the peak and the transcription start site (TSS). BETA ranks genes on the basis of the product between[1]: the regulatory potential of factor binding, using a monotonically decreasing function that is based on the distance between the binding site and the TSS, and[2] differential expression upon factor binding. BETA

then tests the cumulative distribution function of the up and down-regulated genes using a background of non-differentially expressed genes and a one-tailed Kolmogorov-Smirnov test.

**Transcription factor enrichment analysis**. Transcription factor enrichment analysis was performed using the Overrepresentation function of the Genomatix Genome Analyzer[79]. The overrepresentation results are given as a Z-score, which represents the distance from the population mean in units of the population standard deviation. Public ChIP-seq data[33] was downloaded from the SRA database (accessions: SRR5944061, SRR5944062, SRR5944081) and peak calling was performed using the UTAP pipeline. 17,980 peaks that overlapped between replicates were analyzed and compared to genomic background. TF enrichment analysis was also performed on the promoters of the R273 signature genes (145 sequences) compared to the promoters of all canonical genes from the MARS-seq analysis (60519 non-redundant sequences). Promoter DNA was extracted from the UCSC Table Browser[80], human genome build GRCh38, using the knownCanonical table. 500 bp upstream and 100 bp downstream of the transcription start site was taken.

**Analysis of enrichment of CpG islands in promoters**. CpG islands in promoters of the R273 gene signature were compared to all canonical gene promoters (as described in Transcription factor enrichment analysis). Calculation of overlap between CpG islands and promoters was done using bedtools intersect (version 2.25.0); a minimum of 1 bp overlap between CpG island and the promoter was considered positive. Chi square analysis was used as the statistical test.

**Cell cycle profiling**. Cells were grown in 6 cm dishes for 24 h, trypsinized, and subjected to cell cycle analysis with a Phase-Flow BrdU Cell Proliferation Kit (BioLegend). Briefly, cells were incubated with BrdU for 75 min and labeled with Alexa Fluor-647-conjugated anti-BrdU antibody. Total DNA was stained with DAPI. Then, 50,000 cells were collected and analyzed by multispectral imaging flow cytometry. The percentage of cells in each cell cycle phase was manually determined on the basis of BrdU intensity and total DNA content, using FlowJo (Becton, Dickinson and Company).

**Statistical data analysis**. Independent biological replicates were performed and group comparisons were done as detailed in the figure legends. P values below 0.05 were considered significant. Statistical analysis was performed using the Graph-Pad Prism 9.1.0 software.

**Reporting summary**. Further information on research design is available in the Nature Research Reporting Summary linked to this article.

## Data availability

All sequencing data generated in this study have been deposited in the National Center for Biotechnology Information Gene Expression Omnibus (GEO) and are accessible through GEO Series Accession Number GSE173364. All the other data are available within the article and it's Supplementary Information.

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

## Acknowledgements

We thank Benjamin Geiger for inspiring scientific discussions. We thank Carine Joubran for experimental help and Ron Rotkopf for statistical help. RNA-seq analysis was done with critical advice from Michal Pearl and Hadas Keren-Shaul of the Crown Genomics Institute of the Nancy and Stephen Grand Israel National Center for Personalized Medicine, Weizmann Institute of science. This work was supported in part by the Dr. Miriam and Sheldon G. Adelson Medical Research Foundation, a Center of Excellence grant No. 3165/20 from the Israel Science Foundation, the Robert Bosch Stiftung and the Berthold Leibinger Stiftung, the Thompson Family Foundation, and a grant from Anat and Amnon Shashua, and the Moross Integrated Cancer Center. M.O. is incumbent of the Andre Lwoff chair in molecular biology. Cartoons in Figs. 2a, 3a, and 6a, b were created with BioRender.com.

## Author contributions

O.H. and M.O. designed research; O.H. performed research; N.B., M.S., G.F., S.M., M.M., A.A., O.I., G.M., A.G. and I.G. helped with the experiments; E.F., O.G. and S.B.-D. helped with the analyses; R.Y., Y.A., A.H., G.B., D.K., Y.Y. and M.O. supervised research; O.H., Y.A. and M.O. wrote the paper. All authors discussed the results and commented on the manuscript.

## Competing interests

The authors declare no competing interests.
