## [Peer Review File · Nature Communications]

Different hotspot p53 mutants exert distinct phenotypes and predict outcome of colorectal cancer patientsEditorial Note: Parts of this Peer Review File have been redacted as indicated to remove third-party material where no permission to publish could be obtained.

REVIEWER COMMENTS

Reviewer #1 (Remarks to the Author):

Persistent expression of missense mutants of the tumor suppressor p53 is commonly found in human cancers. Originally thought to be loss-of-function there is growing evidence that these mutants have oncogenic activity and that distinct mutants may behave differently. In this study, the authors compare two common mutants in colorectal cancer, R273H and R175H. They present data to argue that R273H exerts a unique transcriptional effect not seen with p53 null or R175H genotypes and that this contributes to a more aggressive, metastatic tumor phenotype.

The notion of mutant p53 "gain of function" has gained traction and has become quite compelling. An intriguing aspect of this is the idea that mutants have distinct activities but this latter idea has been underexplored. In this manuscript, the author address this significant and innovative concept directly. Thus the focus of the study is quite relevant to the readership of Nature Communications. The finding that 273H has a distinct transcriptome from p53 null or 175H and that this correlates with a more aggressive tumor phenotype is an important one. The presented data largely support this conclusion in a manuscript that is thoughtfully written. There are however several key issues that need to be addressed before the study is suitable for publication.

First, a key and central finding of the manuscript is that one mutant p53 273H not another 175H has this specific transcriptional signature and accompanying effects on metastasis. The authors use knockdown of p53 in a cell line expressing endogenous 273H to show this (as well ectopic expression of the mutants in various other cell lines. What is missing is use of a cell line with 175H endogenous expression and showing that this transcriptional signature is not present when this cell line is compared to a p53 knockdown derivative.

Second, a key control is to show that levels of expression of 273H and 175H are comparable. Although representative immunoblots are shown, they should be quantitated and subjected to statistical analysis (for example, figures 2b, 2e, 3a; actually figures 2e and 3a look like there may be more 273H protein which raises a concern about the qualitative difference between the two mutants).

Third, some of the transcriptional effects are mild although statistically significant (Fig 6d is an important example of this). Might the authors identify other targets in which the fold-differences are greater? In any case, they should indicate fold-difference where possible (again for example Fig 6d).

Fourth, greater details need to be provided for the establishment of the cell lines in Figure 2a. The key issue is whether knockdown of the mutant p53 affected cell proliferation or the establishment of the cells that were chosen for further analysis and whether there might have been selective pressure on the production of the knockout cells.

Fifth, comparisons between 175H expressing cells versus p53 null cells should be provided and with greater detail to show whether there is any 175H-driven transcription in Figure 2d.

Finally, the list of genes in the 273H signature should be included, perhaps as supplemental material. In particular the names of the genes that are the basis for Figure 2d should be shown.

Reviewer #2 (Remarks to the Author):

In this interesting paper, the authors demonstrate that different TP53 missense mutations contribute differently to cancer progression and that the p53R273H mutant has distinct GOF activities in CRC. It is well-established that mutations in p53 occur during the later stages of CRC but the function of specific mutant p53 proteins during these late stages of CRC was not studied. The authors show compared the impact of the two most common hotspot p53 mutants R175H and R173H and interestingly found marked differences between the effects of these two proteins.

Through a series of elegant experiments, they show that p53R273H but not p53R175H control a unique transcriptional program to drive oncogenic pathways involved in more aggressive disease in CRC. Overall, this is an elegant and high impact paper that should be published in Nat Comm. There are a few minor comments that should be addressed.

1. In Figure 1b, can the authors show the ratio of R175H/R273H for stage 1, 2, 3 and 4 instead of combining 1,2 and 3,4?
2. Figure 2a, please perform RT-qPCR to show comparable expression of R273H and R175H. Also, how does the expression of R175H and R273H compare to endogenous p53 in SW480 cells? A Western blot should be performed to address this question.
3. The ChIP experiments performed in Figure 6 show that R273H genes associated with cancer progression. However, it is not clear if the p53R273H protein associates with the DNA of ITGA7 directly or through adaptor proteins. Can the authors comment on this? Can they perform in vitro experiments with say purified p53R273H protein and the ITGA7 RE to determine if this interaction is direct? This is important because it is believed that mutant p53 proteins have lost their ability to bind to DNA.
4. p-values are missing for the data in Figure 4g.

Reviewer #3 (Remarks to the Author):

The present study compares the gain-of-function impact of the two most common hotspot TP53 mutations in CRC, R175H and R273H. The authors found marked differences between the effects of these two mutants in invasive and migratory properties that correlates with survival. p53 R273H but not R175H can orchestrate an aggressive '273 transcriptional signature' associated with increased invasion, metastasis and shorter patient survival. This signature correlates with CRC consensus molecular subtype 4 and is in part due to Rho-signaling driven cytoskeletal dynamics. However, this paper, while interesting, has only limited novelty both in concept and in methodology. Previously, a number of mutant p53 genotype-phenotype correlation studies in patient tumor cohorts, knockin mouse models and cell line models had clearly worked out differential behavior among p53 missense mutant variants including their impact on differential patient survival. Likewise, the idea of exploiting their differential properties to individualize patient treatment is not new (e.g. Sabapathy, K., Lane, D.. *Nat Rev Clin Oncol* 15, 13–30 (2018); Robles AI, Jen J, Harris CC. *Cold Spring Harb Perspect Med.* 2016;6(9):a026294). The current paper adds incremental support that individual missense mutations even among the hotspot subgroup carry differential biological phenotypes and that determining the precise p53 mutation in clinical tumor samples might better guide future CRC patient treatment. Another issue is citation, both omission of important citations and overreliance on very old papers from the 1990s.

Major concerns:

1. The study uses engineered CRC cell line models in culture and in transplantation assays of immunodeficient NSG mice. However, all studies were done with simple ectopically overexpressing R175H and R273H cell lines (via plasmid or lentiviral infections) after endogenous p53 knockout or knockdown. Since the TP53 copy number is essential to study p53 mutant GOF effects, such overexpression studies are not very compelling in this day and age. A CRISPR/Cas9-based knock-in strategy would have been state-of-the art.
2. Also the paper heavily relies on SW480 cells. The choice of SW480 with its trisomic homozygous dual TP53 mutation R273H/P309S in the same allele confounds the interpretation. Alternative CRC cell lines with pure R273H mutations such as HT-29, CZ-1 and HRA19 after CRISPR knockin would still have been a better choice (see Liu Y, Bodmer WF. *PNAS* 2006; PMID: 16418264), allowing endogenous R273H analysis and avoiding engineered KO/kd with ectopic overexpression. Related to that, Lines 202-203 state "As seen in Fig S2c, all tested genes were significantly downregulated in the knockout cells, consistent with their being positively regulated by R273H". This is an overstatement since the observed positive regulation of the indicated genes could be due to the second site P309S mutation. To provide rigorous prove, a derivative SW480 KO line expressing only the P309S mutant should not show upregulation of these genes compared to KO. Can the current SW480 results at least be confirmed by HT-29 cells? The same limitation holds true for the GSEA analysis of the 'R273 signature' of SW480 cells +/- shp53 from published data (Ref 23) in Fig S2d. Also, labeling of Fig S2d is incorrect, instead of

'endogenous R273H', it should say 'SW480' or 'endogenous R273H/P309S'. Moreover, is Ref 25 in text correct? Or is it Ref 23 as in Figure legend?

The same holds for engineered RKO, HCT116 and COLO-205 cells.

3. The p53 R248 missense mutation is essentially as frequent in CRC as R273. Directly relevant here, previous studies in mouse models and in clinical CRC and other tumor cohorts also show increased invasion and poorer survival but were cited (Schulz-Heddergott, PMID: 30107178).

How do the R273H and R175 p53 mutants functionally compare to the R248 Q/W mutants in CRC?

4. The fact that a significant proportion of CRC tumors in Suppl Table 1 are R273C rather than R273H is getting lost in the current Abstract. Is there a functional difference between 273H and 273C in survival, metastasis etc.? Fig S6b, c should have tested this side-by-side by qRT-PCR.

5. Throughout the paper many of the cited references are very old, often from the early 1990s. At least some should be updated or complemented by more recent references.

6. Results:

- Fig. 1: Disease-specific survival in patients with metastatic disease (stage 4, Fig S1c) shows no differential between 273 and 175 mutants. This is quite surprising, given that 273 mutants are so strongly differentially associated with uncommon (unresectable) and multiple metastases (by a factor of 2-fold and nearly 4-fold, respectively). Thus, one could expect that this differential is still maintained or even more pronounced in stage 4. This is not consistent with the given explanation that "this is consistent with the notion that the main effect of R273 mutations is on the rate of progression from early stage CRC to advanced disease".

Related to this, which stages are analyzed in Fig 1e? stages 1-3 only?

- Fig. 2a: The generation of the CRISPR/Cas9-based SW480 derivative line is not explained in Material & Methods. It is also not clear how RKO TP53 KO and HCT116 TP53 KO cells were generated. Is the p53 gene signature down-regulated after p53 KO?

- Fig. 2 and Fig S2, Fig 3 and Fig S3, Fig 4c-f: Parental controls are missing. All analyses should be done in comparison to parental unmanipulated SW480, RKO and HCT116 cells as important controls. Are there differences between e.g. the parental SW480 line and the SW480-R273H line in e.g. the 'R273H signature' by GSEA or in migration (Fig. 4c, e)? And it should be confirmed that parental SW480 show GOF (parental SW480 vs SW480-KO).

- Fig. 2d: The list of genes of the 'R273 signature' should be presented as Supplementary table.

- line 201 – 203: Which Figure represents the statement "... with their being positively regulated by p53R273H"?

- Fig. 2c and S2a: A detailed overlay of the 2 heatmaps from Fig. 2c and Fig. S2a should be done to compare both cell systems. How well do they agree?

Again, in Fig. S2a the RNA-seq data of the shc control group (as seen in Figure 2e) is missing.

- Fig. 3, Fig S2 and Fig S3: A broader panel of genes from the R273 signature should be tested by qRT-PCR.

- Fig. 3d: Which genes are regulated in R273 CRC tumors? This information should be provided in a Suppl table.

- Generally, all red/blue GSEA plots testing the R273 signature gene set in cell lines and tumors (Fig 2f, g; Fig 3d, e; Fig S2d Fig S4b, c; Fig S5b) are plotted as negative NES scores. While correct, this is confusing to readers less familiar with GSEA plotting. It would be helpful to switch all comparisons with R273H overexpressing samples on the left and references on the right, producing positive NES scores.

- Fig. 4c: Migration and invasion assays should also be performed in the HCT116 and COLO205 cell set from Supp 3a and c.

- Fig S5b: ..." RNA-seq analysis six hours after plating (Fig. S5b) showed that already at this early time point the R273 signature was upregulated in the p53R273H expressors to a similar extent as after 24 hours. This supports the notion that the inherent gene expression pattern dictated by

p53R273H drives cell spreading, rather than being secondary to it". This is a misleading statement. The system used here is not inducible - which would have provided stronger evidence for causality – but instead KO is compared to stable OE. So no surprise that these cells express the 273 signature. Thus, it does not make sense to speak of " already at this early timepoint...". Also, Fig S5b, Line 281 states " to a similar extent as after 24 hours ". Where is the 24h GSEA shown?

- Fig 6: only generic "mutp53 chromatin binding peaks" were analyzed. No definition given whether they are promoter or enhancer regions and no definition of the chosen peak-calling regions with distance to transcription start and stop sites. Since this binding is most likely indirect (as stated), no attempt to identify at least some mediating partner candidate transcription factors was made.

- Fig. 6d and S6a,c: All 4 conditions (EV, R273H, R273H TAD and R273C) should be directly compared side-by-side within the same experiment to yield a direct comparison between groups. The 273-mediated induction of shown genes ,which are presumably among the best responders, are modest.

Minor

The mp4 movie files are confusing, not properly labeled for cell identity. What is what ?

Figs S1b and 1c: add patient numbers

Line 154, Fig 1f: incorrect statement, sex matters. Being male is a risk factor for R273 CRC tumors.

Fig. 2e, f, line 194-197: What exactly do the authors mean with "... the 'R273 signature', deduced from the reconstituted p53KO cells, was strongly correlated with the differences in gene expression between the R273H reconstituted shp53 cells and the control ..."? Do you mean cells from Fig. 2a (CRISPS-Cas9 KO cells), or from Fig. S2b (sh-p53 cells)? The term p53KO cells is misleading here if they mean the shp53 SW480 cells.

Fig. 3c: Which R273H cell lines were exactly used? Include in legend.

Fig 4g: Statistical significance and method used missing.

Lines 175/ 176: p53 mutation in SW480 is not properly described. " DNA sequence analysis showed that all three p53 copies in SW480 carry two point mutations (R273H and P309S)". Ref 23 is not the most appropriate Ref for SW480. <https://doi.org/10.1016/j.jmb.2005.06.033> is preferred.

Ref 13: Olive et al, Cell 2004: really mainly osteosarcoma rather than T-lymphoma ?

Ref 14: incorrect claim from obscure journal: Colorectal cancer (CRC) is the third most common cancer worldwide and the fourth most common cause of cancer death (Int J Cancer 2019 Jun 15;144(12):2992-3000 PMID: 30536395).

We would like to thank the reviewers for their insightful comments. We believe that the additional experiments and analyses that we carried out in response to those comments have improved the paper significantly.

Please find below our point-by-point response to the reviewer's comments. The original comments are in regular font, and our replies are in italics.

In the enclosed manuscript file, the new text that was added in order to address the reviewers' comments is in red font.

Reviewer #1 (Remarks to the Author):

Persistent expression of missense mutants of the tumor suppressor p53 is commonly found in human cancers. Originally thought to be loss-of-function there is growing evidence that these mutants have oncogenic activity and that distinct mutants may behave differently. In this study, the authors compare two common mutants in colorectal cancer, R273H and R175H. They present data to argue that R273H exerts a unique transcriptional effect not seen with p53 null or R175H genotypes and that this contributes to a more aggressive, metastatic tumor phenotype.

The notion of mutant p53 "gain of function" has gained traction and has become quite compelling. An intriguing aspect of this is the idea that mutants have distinct activities but this latter idea has been underexplored. In this manuscript, the author address this significant and innovative concept directly. Thus the focus of the study is quite relevant to the readership of Nature Communications. The finding that 273H has a distinct transcriptome from p53 null or 175H and that this correlates with a more aggressive tumor phenotype is an important one. The presented data largely support this conclusion in a manuscript that is thoughtfully written. There are however several key issues that need to be addressed before the study is suitable for publication. First, a key and central finding of the manuscript is that one mutant p53 273H not another 175H has this specific transcriptional signature and accompanying effects on metastasis. The authors use knockdown of p53 in a cell line expressing endogenous 273H to show this (as well ectopic expression of the mutants in various other cell lines. What is missing is use of a cell line with 175H endogenous expression and showing that this transcriptional signature is not present when this cell line is compared to a p53 knockdown derivative.

To answer this important issue, we conducted a comparison between two additional CRC cell lines before and after mutp53 knockdown. One of those cell lines (HT-29) expresses endogenous R273H while the other (COGA-5) expresses endogenous R175H (https://web.expasy.org/cellosaurus/CVCL_A076). We then compared the mRNA levels of representative R273 signature genes. While knockdown of endogenous R273H in HT-29 significantly downregulated most of the tested genes, knockdown of endogenous R175H in COGA-5 did not affect significantly any of those genes. These results are now included in figure 3d-g of the revised manuscript.

Incidentally, we spent much time and effort in attempts to obtain CRC cell lines expressing endogenous R175H from at least 3 different labs. This turned out to be more challenging than we had expected, and has made us realize that although R175H is the most common single TP53 mutation in CRC, lines expressing it are disproportionately scarce, as compared to the abundance of R273 and R248 mutant CRC lines. Moreover, those R175 lines that do exist are hard to grow and take much time to recover fully from freezing. We believe that this is probably not a mere coincidence, and it may attest to the greater ability of R273 and R248-mutated CRC cells to cope with stressful conditions, which may also account for their increased ability to transition successfully from the primary tumor site to metastatic sites.

Second, a key control is to show that levels of expression of 273H and 175H are comparable. Although representative immunoblots are shown, they should be quantitated and subjected to statistical analysis (for example, figures 2b, 2e, 3a; actually figures 2e and 3a look like there may be more 273H protein which raises a concern about the qualitative difference between the two mutants).

We have now quantified the immunoblots from the different cell lines. Overall, this showed that the levels of the R273H and R175H proteins were not significantly different in any of the cell lines (see Reviewer Fig. 1). Several representative examples are now included in the revised MS (Fig. 2b, Fig. S2c, Fig. S3a, Fig. S3b, S3d). Moreover, we compared the transcriptional output of genes from the R273H signature between the parental SW480 cells (expressing endogenous R273H/P309S) and the SW480 R175H overexpressing cells. As expected, the amount of R175H protein was several fold higher than that of the endogenous R273H/P309S in the parental cells (Fig. S2c). Nevertheless, as shown in figure S2d, the expression of R273 genes was significantly higher in the parental cells, despite the lower amount of mutp53 protein. Overall, this indicates that the differential effects of R273H, as compared to R175H, are not due to higher abundance of R273H.

Reviewer Fig1. Quantification of Western blot data from two biological repeats, using Image Lab (Bio-Rad). Statistical analysis was done using unpaired two-tailed t-test.

Third, some of the transcriptional effects are mild although statistically significant (Fig 6d is an important example of this). Might the authors identify other targets in which the fold-differences are greater? In any case, they should indicate fold-difference where possible (again for example Fig 6d).

We agree with the reviewer that the effects in Fig. 6d (Fig. 7d in the revised MS) are mild. However, it is important to point out that the experiment depicted in that figure is a transient transfection experiment, where only a minority of the cells in the culture were successfully transfected. Plausibly, the effect of R273H in those positive cells was diluted by the unaltered expression of the tested genes in the rest of the cells in the culture. In contrast, the transcriptional effect of R273H when the entire population is stably expressing this mutant is quite robust, as shown for example in Fig. S2d and Fig. S5b of the revised manuscript. Still, it should be mentioned that although in the transient transfection experiments the effects were indeed mild, as pointed out by the reviewer, they were very consistent in all five biological replicates of these experiments.

Fourth, greater details need to be provided for the establishment of the cell lines in Figure 2a. The key issue is whether knockdown of the mutant p53 affected cell proliferation or the establishment of the cells that were chosen for further analysis and whether there might have been selective pressure on the production of the knockout cells.

The p53 knockout cell lines used in this study (SW480 and RKO) were obtained from collaborators and not generated by us. We have now added the relevant references, describing their establishment. Also, to address the reviewer's comment, we tested experimentally if there is any difference in proliferation rates between the parental SW480 cells and our SW480 p53 knockdown cells. As can be seen in Reviewer Fig. 2, we did not notice a significant difference.

Reviewer Fig 2. Proliferation assay of SW480 shControl (shc) and shp53 cells. 100k cells were plated in a 6-well dish and cells were counted after 72 hours using a hemocytometer.

Fifth, comparisons between 175H expressing cells versus p53 null cells should be provided and with greater detail to show whether there is any 175H-driven transcription in Figure 2d.

To address this important comment, we have now included in the revised manuscript volcano plots comparing the RNA-seq data of the SW480 KO cells to either the R175H or the R273H overexpressors. As can be seen in Fig S2a, while there was a strong effect of R273H overexpression on the transcriptome (544 upregulated genes and 486 downregulated genes; p-value<0.05, Fold Change >1.5), only a total of 73 genes were differentially expressed between the R175 overexpressors and the knockout cells. This was quite surprising, as R175H has been shown to exert significant phenotypic and transcriptional effects in a variety of other settings. Yet, this same mutant has only minimal transcriptional and phenotypic effects within the particular background of SW480 cells. Of note, we did observe measurable phenotypic effects of R175H in other cell lines in our study, such as COLO-205 and HCT116 CRISPR/Cas9 knock-in, as shown now in the revised MS. This underscores the notion that the gain-of-function effects of different p53 mutants in cancer are greatly context-dependent.

Finally, the list of genes in the 273H signature should be included, perhaps as supplemental material. In particular the names of the genes that are the basis for Figure 2d should be shown.

We have added a supplementary table (supplementary table 3) with the R273 signature genes.

Reviewer #2 (Remarks to the Author):

In this interesting paper, the authors demonstrate that different TP53 missense mutations contribute differently to cancer progression and that the p53R273H mutant has distinct GOF activities in CRC. It is well-established that mutations in p53 occur during the later stages of CRC but the function of specific mutant p53 proteins during

these late stages of CRC was not studied. The authors show compared the impact of the two most common hotspot p53 mutants R175H and R173H and interestingly found marked differences between the effects of these two proteins. Through a series of elegant experiments, they show that p53R273H but not p53R175H control a unique transcriptional program to drive oncogenic pathways involved in more aggressive disease in CRC. Overall, this is an elegant and high impact paper that should be published in Nat Comm. There are a few minor comments that should be addressed.

1. In Figure 1b, can the authors show the ratio of R175H/R273H for stage 1, 2, 3 and 4 instead of combining 1,2 and 3,4?

We performed the analysis requested by the reviewer. The result is shown in Reviewer Fig. 3. As seen, the decrease in the ratio of R175 mutant tumors to R273 mutant tumors is even more pronounced in stage 4 tumors than it is in Stage 3 tumors. However, owing to the reduced number of samples in each bin, the differences do not reach statistical significance. Statistical significance is reached only when we pool the samples into 2 groups, as done in the original figure. Therefore, we prefer to keep Figure 1b as is.

Reviewer Fig. 3. Ratio between the numbers of CRC cases with R175 mutations and R273 mutations in stage 1,2,3 and 4.

2. Figure 2a, please perform RT-qPCR to show comparable expression of R273H and R175H. Also, how does the expression of R175H and R273H compare to endogenous p53 in SW480 cells? A Western blot should be performed to address this question.

To address the reviewer's request, we performed RT-qPCR to compare the mRNA levels of R175H and R273H in SW480 overexpressing cells. As seen in reviewer figure 4 (left), the R175H mRNA levels are actually higher than those of the R273H mRNA, despite the fact that the mutant p53 protein levels are comparable. Although the difference did not reach statistical significance, we saw a similar trend in multiple cell lines. This suggests that the R273 mutant proteins may have a somewhat longer half-life. Interestingly, when we examined TP53 mRNA levels in TCGA CRC tumors (reviewer fig 4, right) we observed that tumors harboring R175 mutations express significantly higher p53 mRNA levels when compared to tumors harboring R273H mutations.

Reviewer Fig 4. Left: TP53 mRNA levels in control SW480 KO cells (KO) and in SW480 KO cells stably overexpressing either R175H or R273H. Statistical analysis was done using one-way ANOVA and Tukey's post-hoc test. right: TP53 mRNA levels in CRC tumors harboring either R175H or R273H mutations. Data from TCGA COAD-READ cohort. Wilcoxon test.

Regarding the second comment, we have now added a Western blot comparing the mutant p53 levels in the parental SW480 cell line and in the SW480 KO cells overexpressing either R175H or R273H. This is included as Fig S2c of the revised manuscript. As can be seen in this figure, the overexpression is relatively modest, reaching p53 levels that are about 3 fold above those of the endogenous mutant p53. Such levels are still within the range observed for the endogenous p53 in some mutant p53 cancer cell lines.

3. The ChIP experiments performed in Figure 6 show that R273H genes associated with cancer progression. However, it is not clear if the p53R273H protein associates with the DNA of ITGA7 directly or through adaptor proteins. Can the authors comment on this? Can they perform in vitro experiments with say purified p53R273H protein and the ITGA7 RE to determine if this interaction is direct? This is important because it is believed that mutant p53 proteins have lost their ability to bind to DNA.

We thank the reviewer for this important comment. Like the reviewer, and in agreement with a large body of published studies, we too believe that the association of R273H with the DNA of R273 signature gene is indirect and occurs through piggybacking of the mutant p53 protein to a variety of sequence-specific transcription factors. In the revised manuscript (Fig 7e), we have now included a list of transcription factors that are predicted to bind preferentially to R273 signature gene promoters and are also enriched in mutp53 ChIP-seq peaks of SW480 cells, and may thus serve as an “anchor” for R273H. We apologize that this point was not clearly explained in the previous version of the manuscript, and have now modified the text accordingly.

4. p-values are missing for the data in Figure 4g.

We added p-values for Figure 4g (figure 4h in the revised manuscript).

Reviewer #3 (Remarks to the Author):

The present study compares the gain-of-function impact of the two most common hotspot TP53 mutations in CRC, R175H and R273H. The authors found marked differences between the effects of these two mutants in invasive and migratory properties that correlates with survival. p53 R273H but not R175H can orchestrate an aggressive ‘273 transcriptional signature’ associated with increased invasion, metastasis and shorter patient survival. This signature correlates with CRC consensus molecular subtype 4 and is in part due to Rho-signaling driven cytoskeletal dynamics.

However, this paper, while interesting, has only limited novelty both in concept and in methodology. Previously, a number of mutant p53 genotype-phenotype correlation studies in patient tumor cohorts, knockin mouse models and cell line models had clearly worked out differential behavior among p53 missense mutant variants including their impact on differential patient survival. Likewise, the idea of exploiting their differential properties to individualize patient treatment is not new (e.g. Sabapathy, K., Lane, D.. Nat Rev Clin Oncol 15, 13–30 (2018); Robles AI, Jen J, Harris CC. Cold Spring Harb Perspect Med. 2016;6(9):a026294). The current paper adds incremental support that individual missense mutations even among the hotspot subgroup carry differential biological phenotypes and that determining the precise p53 mutation in clinical tumor samples might better guide future CRC patient treatment. Another issue is citation, both omission of important citations and overreliance on very old papers from the 1990s.

Major concerns:

1. The study uses engineered CRC cell line models in culture and in transplantation assays of immunodeficient NSG mice. However, all studies were done with simple ectopically overexpressing R175H and R273H cell lines (via plasmid or lentiviral infections) after endogenous p53 knockout or knockdown. Since the TP53 copy number is essential to study p53 mutant GOF effects, such overexpression studies are not very

compelling in this day and age. A CRISPR/Cas9-based knock-in strategy would have been state-of-the art.

We thank the reviewer for this critical comment. As suggested by the reviewer, we have now employed a CRISPR/Cas9-based knock-in strategy to introduce different p53 mutants endogenously. To that end, we knocked in either R175H or R273H into the p53 wild type (WT) cell line HCT116, using state-of-the-art RNP CRISPR knock-in methodology (1)(Fig 3a of the revised manuscript). The homozygous mutations were validated using Sanger sequencing, and WT p53 loss of function was confirmed by measuring p21 mRNA levels (Fig 3b). To reduce the effects of variations among individual single cell-derived clones, we obtained 5 independent cell clones for each mutant, and then subjected each of them to RT-qPCR analysis and averaged the data from all 5 clones carrying the same p53 mutant. As seen in Fig 3c, the R273H knock-in clones significantly upregulated the mRNA levels of R273H signature genes, relative to cells retaining WT p53 (Parental HCT116 and CRISPR control cells) and to the HCT116 R175H mutant clones. In contrast, these genes were upregulated only mildly, or not at all, in the R175H knock-in cells.

In addition, in line with the overexpression data, R273H clones showed a significant increase in migratory and invasive capacity in transwell migration/invasion assays, relative to the control WT p53 cells (Fig 5e-g).

Altogether, this new data validates our earlier conclusions, obtained with ectopic mutant p53, also for endogenously expressed mutant p53.

2. Also the paper heavily relies on SW480 cells. The choice of SW480 with its trisomic homozygous dual TP53 mutation R273H/P309S in the same allele confounds the interpretation. Alternative CRC cell lines with pure R273H mutations such as HT-29, CZ-1 and HRA19 after CRISPR knockin would still have been a better choice (see Liu Y, Bodmer WF. PNAS 2006; PMID: 16418264), allowing endogenous R273H analysis and avoiding engineered KO/kd with ectopic overexpression.

Related to that, Lines 202-203 state “As seen in Fig S2c, all tested genes were significantly downregulated in the knockout cells, consistent with their being positively regulated by R273H”. This is an overstatement since the observed positive regulation of the indicated genes could be due to the second site P309S mutation. To provide rigorous prove, a derivative SW480 KO line expressing only the P309S mutant should not show upregulation of these genes compared to KO. Can the current SW480 results at least be confirmed by HT-29 cells?

The same limitation holds true for the GSEA analysis of the ‘ R273 signature’ of SW480 cells +/- shp53 from published data (Ref 23) in Fig S2d. Also, labeling of Fig S2d is incorrect, instead of ‘endogenous R273H’, it should say ‘SW480’ or ‘endogenous R273H/P309S’. Moreover, is Ref 25 in text correct? Or is it Ref 23 as in Figure legend? The same holds for engineered RKO, HCT116 and COLO-205 cells.

We agree with the reviewer's comment about the complexity of the TP53 mutational status of the SW480 cells. It is important to mention, though, that while the overexpression model has its limitation, it allows the comparison of two different mutants expressed at similar levels. In addition, it has the advantage of using the

same vector, unlike the CRISPR/Cas9 system, which employs different gRNAs for generation of the different mutants and therefore may give rise to differential off-target effects.

Indeed, while one might formally argue that the effects seen in SW480 cells are not necessarily due only to the R273H mutant, our overexpression models show clearly that the R273 mutant is a potent driver of the R273 signature. This is also supported by the analysis of data from TCGA, where the vast majority of R273 mutated tumors do not express an additional mutation. Yet, to address more directly the reviewer's question about the impact of endogenous R273-mutated p53 in CRC cells, we have followed the reviewer's suggestion and established CRISPR knock-in derivatives of HCT116 cells. The choice to perform the knock-in in wtp53-expressing HCT116 cells, rather than in TP53 mutated cells, was guided by practicality considerations: knock-in of R175H into a cell line endogenously harboring a R273 mutation would have required to first knock in the R175H mutation and then replace the R273H mutation by a wt sequence, rendering the process quite cumbersome. Moreover, using 2 different guide RNA sequences on the same cells may potentially double the chances for off-target effects. Reassuringly, the analysis of the HCT116-derived knock-in cells confirmed that introduction of endogenous R273H recapitulates the transcriptional and phenotypic effects (migration and invasion) observed previously upon R273H overexpression.

Moreover, as requested by the reviewer, we knocked down the endogenous R273H in HT-29 cells, and compared the effects of such knockdown to the effects of knocking down the endogenous R175H in COGA-5 cells (https://web.expasy.org/cellosaurus/CVCL_A076). As shown in figure 3d-g, while knocking down R273H in HT-29 cells impaired significantly the expression of R273 signature genes, this was not seen upon R175H knockdown in COGA-5 cells.

As requested by the reviewer, we also looked into the impact of the P309S mutation. Of note, this is a very rare mutation in cancer: only 6 cases of this mutation are recorded in the entire database of IARC p53 (https://tp53.isb-cgc.org/results_somatic_mutation_list), which strongly suggests that it is a passenger mutation and not a driver mutation. Yet, to address the reviewer's concern more directly, we employed site-directed mutagenesis to generate a P309S-expressing recombinant lentivirus. However, all attempts to stably express the P309S mutant in the SW480 KO cells resulted in cell death. The same outcome was attained when we tried to express wt p53 in these. This suggested that P309S might indeed be just a passenger mutation, which does not compromise the activity of wt p53 and is thus functionally equivalent to a synonymous mutation. To formally test this possibility, we transiently transfected the SW480 KO cells with a plasmid encoding P309S. As can be seen in Reviewer figure 5, P309S indeed upregulated both p21 mRNA and PUMA mRNA to a similar extent as bona fide wt p53. As expected, R273H had no effect on the levels of either p21 mRNA or PUMA mRNA. Hence, we believe that the P309 is just an inconsequential passenger mutation, and thus the double-mutated protein expressed in SW480 cells is functionally very similar to a protein carrying only the R273H mutation.

Regarding the labeling of the SW480 cells in the text and in the GSEA, we have modified the labeling to include both mutations as suggested by the reviewer.

Reviewer Fig 5. mRNA levels of p53, p21 and PUMA in SW480 p53 KO cells after transient transfection of either empty vector or expression plasmids encoding wt p53, p53^{P309S} or p53^{R273H}. Statistical analysis was done using one-way ANOVA and Tukey's post hoc test.

3. The p53 R248 missense mutation is essentially as frequent in CRC as R273. Directly relevant here, previous studies in mouse models and in clinical CRC and other tumor cohorts also show increased invasion and poorer survival but were cited (Schulz-Heddergott, PMID: 30107178). How do the R273H and R175 p53 mutants functionally compare to the R248 Q/W mutants in CRC?

As requested by the reviewer, we interrogated the impact of the 2 hotspot R248 mutations, namely R248Q and R248W, in CRC. To that end, we employed site-directed mutagenesis to generate recombinant lentiviruses expressing either R248Q or R248W, and stably overexpressed them in SW480 p53 KO cells. As can be seen in new Fig S5a, the expression levels of those mutants were similar to that of R273H. Importantly, both R248Q and R248W significantly upregulated genes derived from the R273H signature, to a similar extent as R273H (new Fig. S5b). Moreover, GSEA analysis indicated that R248-mutated CRC tumors are also selectively associated with increased expression of the R273 signature, when compared to either R175H tumors or p53-truncated tumors (new Fig S5c-d). Remarkably, the disease-specific survival of patients with R248-mutated CRC tumors was very similar to that of patients with R273 mutations, and significantly shorter than of those with R175 mutated tumors

(new Fig S5f). Thus, the impact of R248 mutations is very similar to that of R273 mutations. A mention of the R248 observations has also been added to the Abstract. We thank the reviewer for bringing up this clinically relevant issue, and believe that our new data broadens the scope of this study.

4. The fact that a significant proportion of CRC tumors in Suppl Table 1 are R273C rather than R273H is getting lost in the current Abstract. Is there a functional difference between 273H and 273C in survival, metastasis etc.? Fig S6b, c should have tested this side-by-side by qRT-PCR.

We added a mention of 273C to the revised Abstract.

To address the reviewer's question, we examined the available data to see whether there are any differences between the R273C and R273H mutant tumors, and did not observe any significant differences in relative abundance of multiple metastases or uncommon metastases. Similarly, we could not detect any significant difference in disease specific overall survival. This is now shown in Fig S7d. We also modified the original Fig. S6b,c (now Fig. S7) to show the data side-by-side with the Western blot and the RT-qPCR.

5. Throughout the paper many of the cited references are very old, often from the early 1990s. At least some should be updated or complemented by more recent references.

As suggested, we added several recent references to complement the papers from the 1990s, which we would still like to leave in the bibliography owing to their fundamental contributions to our current knowledge.

6. Results:- Fig. 1: Disease-specific survival in patients with metastatic disease (stage 4, Fig S1c) shows no differential between 273 and 175 mutants. This is quite surprising, given that 273 mutants are so strongly differentially associated with uncommon (unresectable) and multiple metastases (by a factor of 2-fold and nearly 4-fold, respectively). Thus, one could expect that this differential is still maintained or even more pronounced in stage 4. This is not consistent with the given explanation that "this is consistent with the notion that the main effect of R273 mutations is on the rate of progression from early stage CRC to advanced disease". Related to this, which stages are analyzed in Fig 1e? stages 1-3 only?

We thank the reviewer for bringing up this issue, which is indeed confusing. Although the differences related to multiple metastases and uncommon metastases are indeed significant, the relative proportion of these patients out of the total metastatic cases is quite low: only about 1/3 of R273 mutant tumors have uncommon metastases and only about 1/5 have multiple metastases. Therefore, the differential

effect of tumors with multiple/uncommon metastases on overall survival is probably diluted and obscured by the majority of samples that do not possess multiple/uncommon metastases. Moreover, the number of stage 4 patients in this analysis was much lower than that of stage 1-3 patients (20 vs 230, respectively). Based on the above, and in light of the feedback from the reviewer, we realized that the original Fig. S1c and 1e were probably more confusing than helpful. We therefore decided to omit it and keep only the analysis shown in the revised Fig 1e, which takes account of all disease stages together.

Fig. 2a: The generation of the CRISPR/Cas9-based SW480 derivative line is not explained in Material & Methods. It is also not clear how RKO TP53 KO and HCT116 TP53 KO cells were generated. Is the p53 gene signature down-regulated after p53 KO?

These knockout lines were not generated by us but were actually obtained from other labs, who have described the generation of those lines in their respective publications. We apologize that this was not clear enough in the previous version, and have now included the relevant references for the SW480 and RKO p53KO cells in the Methods section. Regarding the HCT116 cells, we have now replaced the p53KO-based overexpression experiments with the new experiments the CRISPR/Cas9 knock-in cells that we generated for the revision of this MS. A detailed description of these CRISPR/Cas9 knock-in cells is also included in the Methods section. Down-regulation of p21 expression in the RKO knockout cells, both under basal conditions and after 5-FU treatment, was shown by Sur et al, 2009, and is reproduced below.

[REDACTED]

*From: Sur, S. et al. A panel of isogenic human cancer cells suggests a therapeutic approach for cancers with inactivated p53. Proc. Natl. Acad. Sci. U. S. A. **106**, 3964 (2009).*

- Fig. 2 and Fig S2, Fig 3 and Fig S3, Fig 4c-f: Parental controls are missing. All analyses should be done in comparison to parental unmanipulated SW480, RKO and HCT116 cells as important controls. Are there differences between e.g. the parental

SW480 line and the SW480-R273H line in e.g. the ‘R273H signature’ by GSEA or in migration (Fig. 4c, e)? And it should be confirmed that parental SW480 show GOF (parental SW480 vs SW480-KO).

As requested, we have added parental controls for RKO (figures S3b-c). As for the parental SW480 cells, we have performed RT-qPCR analysis to compare these cells not only to the SW480 KO cells but also to the R175H and R273H overexpressing SW480 cells. In the revised manuscript we show that SW480 parental cells indeed exhibit GOF effects, with regard to both upregulation of R273 signature genes (fig S2d) and migratory capacity (Fig. S6d-e). Perhaps not surprisingly, the impact of the overexpressed R273H tended to be quantitatively greater than that of the endogenous SW480 mutant p53, but both were clearly significant relative to the control p53-KO cells. In contrast, the differences between the overexpressed R273H and the endogenous SW480 mutant p53 were not statistically significant.

- Fig. 2d: The list of genes of the ‘R273 signature’ should be presented as Supplementary table.

We have added a supplementary table (supplementary table 3) with the R273 signature genes.

- line 201 – 203: Which Figure represents the statement “... with their being positively regulated by p53R273H ”?

That statement referred to the old Fig S2c, which showed higher mRNA levels of R273 signature genes in the parental SW480 cells when compared to SW480 knockout cells. However, we have now performed new RT-qPCR analysis of the parental cells, the p53 KO cells and the mutant overexpressing cells, and therefore this statement now refers to fig S2d.

- Fig. 2c and S2a: A detailed overlay of the 2 heatmaps from Fig. 2c and Fig. S2a should be done to compare both cell systems. How well do they agree?

As requested, we have generated a heatmap overlaying both systems (SW480 KO cells and SW480 shp53 cells). As can be seen in the revised manuscript (Fig S2b), there is a very good agreement between the systems with regard to the majority of the R273 signature genes.

Again, in Fig. S2a the RNA-seq data of the shc control group (as seen in Figure 2e) is missing.

We have not performed RNA seq on the shc cells. Our main aim was to identify differentially expressed genes between the different p53 mutants. As already pointed out by this reviewer, comparison with the endogenous p53 of SW480 might have been confounded by the additional presence of the P309S mutant in these cells. Therefore, we preferred to perform a more “clean” comparison, between cells differing only by a single mutant. However, as seen in figure S2d, we have now validated that the control SW480 cells indeed show upregulation of representative genes derived from the RNA-seq, when compared to the p53 KO cells.

- Fig. 3, Fig S2 and Fig S3: A broader panel of genes from the R273 signature should be tested by qRT-PCR.

We have expanded the panel of genes from the R273 signature for all cell lines.

- Fig. 3d: Which genes are regulated in R273 CRC tumors? This information should be provided in a Suppl table.

We have now added supplementary table 4, listing the genes that are upregulated by R273 in CRC tumors, when compared to both R175 tumors and truncated p53 tumors.

- Generally, all red/blue GSEA plots testing the R273 signature gene set in cell lines and tumors (Fig 2f, g; Fig g3d, e; Fig S2dFig S4b, c; Fig S5b) are plotted as negative NES scores. While correct, this is confusing to readers less familiar with GSEA plotting. It would be helpful to switch all comparisons with R273H overexpressing samples on the left and references on the right, producing positive NES scores.

We thank the reviewer for this comment. To avoid confusion we have changed the position of the GSEA plots to produce positive NES scores.

- Fig. 4c: Migration and invasion assays should also be performed in the HCT116 and COLO205 cell set from Supp 3a and c.

We have now used our new HCT116 CRISPR/Cas9 knock-in cells and preformed both migration and invasion assays, comparing the control wild type p53 cells to the

R175H knock-in clones and the R273H knock-in clones. As can be seen in the revised manuscript figure 5e-g, R273H knock-in clones migrated and invaded significantly better than the WT control cells and the R175H knock-in clones.

As requested, we have also conducted a migration assay for the COLO-205 cells and added the results to figure S6h-i. As can be seen, R273H augmented the migratory capacity also in these cells. We also preformed invasion assays in the COLO-205 system; however, we could not detect any invading cells in this system, regardless of p53 status.

- Fig S5b: ...” RNA-seq analysis six hours after plating (Fig. S5b) showed that already at this early time point the R273 signature was upregulated in the p53R273H expressors to a similar extent as after 24 hours. This supports the notion that the inherent gene expression pattern dictated by p53R273H drives cell spreading, rather than being secondary to it”. This is a misleading statement. The system used here is not inducible - which would have provided stronger evidence for causality – but instead KO is compared to stable OE. So no surprise that these cells express the 273 signature. Thus, it does not make sense to speak of “ already at this early timepoint...”.

Also, Fig S5b, Line 281 states “ to a similar extent as after 24 hours “. Where is the 24h GSEA shown?

The 24 hours comparison is the RNA-seq analysis done on the SW480 KO system. This is the data from which we generated the R273 signature; therefore there is no point in the GSEA on these samples, as we will be comparing identical sets of genes. Moreover, we conducted a GSEA-based comparison between the gene expression patterns of the R273-expressing cells 6 hours vs 24 hours after plating, using the R273 gene signature as the tested gene set. As shown in Reviewer fig7, there was no significant difference between the two time points. We thus concluded that the R273 signature is inherent to the R273-overexpressing cells and is not a consequence of cell spreading. We have now changed the wording in page 10 (bottom), to make this point more clearly.

SW80 R273H (6 hours)

SW480 R273 (24 hours)

Reviewer fig7. *GSEA enrichment plot for R273H-overexpressing SW480 cells harvested at different times after plating. The R273H signature was used as the tested gene set.*

- Fig 6: only generic “mutp53 chromatin binding peaks” were analyzed. No definition given whether they are promoter or enhancer regions and no definition of the chosen peak-calling regions with distance to transcription start and stop sites. Since this binding is most likely indirect (as stated), no attempt to identify at least some mediating partner candidate transcription factors was made.

We have added to the Methods section a detailed explanation of the methodology employed for the analyses in fig 7a and 7b (Fig 6 in the original manuscript), describing the regions used for these analyses, including distance to TSS.

To identify putative mediators of the binding of R273H to chromatin, we analyzed the promoter regions of R273H signature genes. This analysis revealed enrichment for binding sites of numerous transcription factors, including SP1 and EGR (Fig 7e). Reassuringly, binding sites for many of those transcription factors are also enriched substantially in mutp53 binding peaks of SW480 cells, determined experimentally by ChIP-seq by another lab. This information is now included in the revised MS. On note, SP1 and EGR1 have been reported to recruit mutp53 to DNA (2,3). However, transient knockdown of SP1 and EGR1 (including double knockdown) did not reduce the expression of representative R273H signature genes in SW480 cells (data not shown), suggesting that either additional transcription factors may mediate the binding of R273H to chromatin, or that R273H may orchestrate long-term epigenetic effects that are not reversible by transient depletion of its DNA-anchoring proteins. This is further supported by the fact that the promoters of R273 signature genes were significantly enriched for association with CpG islands when compared to all promoters (Fig. S7e)

- Fig. 6d and S6a,c: All 4 conditions (EV, R273H, R273H TAD and R273C) should be directly compared side-by-side within the same experiment to yield a direct comparison between groups. The 273-mediated induction of shown genes, which are presumably among the best responders, are modest.

The experiment was done with all 4 conditions together. We have now added a side by side comparison of all 4 groups (fig S7c).

The modest differences are most probably due to the fact that these experiments were done as transient transfections. As estimated by visualization of transfection-derived GFP, the transfection efficiency that we obtained in those cells was around 30-50% in optimal conditions. Therefore, the actual extent of induction is probably 2-3 fold greater than what is shown in the figures. Indeed in the stable expressors, generated

by lentiviral infection, those genes show stronger differences between R273H and KO/R175H, as can be observed in figures S2d and S5b.

Minor

The mp4 movie files are confusing, not properly labeled for cell identity. What is what ?

We have added labeling to the mp4 movies to avoid confusion.

Figs S1b and 1c: add patient numbers

As explained above, we have removed figure S1b and S1c.

Line 154, Fig 1f: incorrect statement, sex matters. Being male is a risk factor for R273 CRC tumors.

We thank the reviewer for bringing up this matter. Indeed, we believe that this may be an interesting example of gender disparity in cancer. Consequently, we have now expanded the presentation of the data showing the gender-specific difference in the impact of R273 mutations on patient outcome, and included it as Fig. S1b,c. along with a short discussion in the text.

Fig. 2e, f, line 194-197: What exactly do the authors mean with "... the 'R273 signature', deduced from the reconstituted p53KO cells, was strongly correlated with the differences in gene expression between the R273H reconstituted shp53 cells and the control ..."? Do you mean cells from Fig. 2a (CRISPS-Cas9 KO cells), or from Fig. S2b (sh-p53 cells)? The term p53KO cells is misleading here if they mean the shp53 SW480 cells.

We apologize if this sentence was not clear enough. The R273 signature was generated from RNA-seq performed on SW480 KO cells reconstituted with R273H, compared to R175H overexpressors and control p53 KO cells. We then performed GSEA to compare this signature to the data obtained from a separate RNA-seq analysis, which compared SW480 shp53 cells (using 3' UTR-directed p53 knock down) to SW480 shp53 cells overexpressing either R175H or R273H. The aim was to test whether the R273 signature, generated in the p53 KO system, is upregulated also in the shp53 system upon R273H overexpression. As seen in Fig 2f-g, our GSEA analysis confirmed that this is indeed the case. We have rephrased the sentence accordingly (page 7, middle).

Fig. 3c: Which R273H cell lines were exactly used? Include in legend

The identities of R273H cell lines were added to the figure legend.

Fig 4g: Statistical significance and method used missing.

Statistical significance and method used were added

Lines 175/ 176: p53 mutation in SW480 is not properly described. “ DNA sequence analysis showed that all three p53 copies in SW480 carry two point mutations (R273H and P309S)”.

We changed the phrasing according to the reviewer’s suggestion, including the relevant reference.

Ref 23 is not the most appropriate Ref for SW480. <https://doi.org/10.1016/j.jmb.2005.06.033> is preferred.

We changed reference 23 as per the reviewer’s suggestion

Ref 13: Olive et al, Cell 2004: really mainly osteosarcoma rather than T-lymphoma ?

In the paper by Olive et al. the R172 mutant demonstrated an increase in osteosarcomas compared to p53 null and p53 R270 mutant. T-cell lymphomas were more common in p53 null mice, as described also in other papers. Nevertheless, we changed the phrasing slightly to match more precisely the original statement.

From the paper:

In contrast to the frequent carcinomas in p53R270H/+ mice, the most frequent tumors in the p53R172H/+ mice were osteosarcomas. Roughly twice as many p53R172H/+ mice developed osteosarcomas as did p53^{+/-} and p53R270H/+ mice (χ^2 , $p = 0.043$ for p53^{+/-} mice, $p = 0.039$ for p53R270H/+ mice)'

Ref 14: incorrect claim from obscure journal: Colorectal cancer (CRC) is the third most common cancer worldwide and the fourth most common cause of cancer death (Int J Cancer 2019 Jun 15;144(12):2992-3000 PMID: 30536395).

Although different reports vary somewhat in the precise ranking of the leading cancers, we note that an update on the global cancer burden using the GLOBOCAN 2020 estimates of cancer incidence and mortality produced by the International Agency for Research on Cancer was published in 2021 in the ACS journal CA Cancer J. Clin. According to that estimate, colorectal is indeed the third most common cancer and the second leading cause of cancer death after lung cancer.

From the paper:

'Female breast cancer has surpassed lung cancer as the most commonly diagnosed cancer, with an estimated 2.3 million new cases (11.7%), followed by lung (11.4%), colorectal (10.0 %), prostate (7.3%), and stomach (5.6%) cancers. Lung cancer remained the leading cause of cancer death, with an estimated 1.8 million deaths (18%), followed by colorectal (9.4%), liver (8.3%), stomach (7.7%), and female breast (6.9%) cancers'

We have replaced the previous reference 14 by the above recent publication .

References

1. Liang, X. *et al.* Rapid and highly efficient mammalian cell engineering via Cas9 protein transfection. *J. Biotechnol.* **208**, 44–53 (2015).
2. Kim, M. P. & Lozano, G. Mutant p53 partners in crime. *Cell Death Differ.* **25**, 161–168 (2018).
3. Liu, J. *et al.* Physical interaction between p53 and primary response gene Egr-1. *Int. J. Oncol.* **18**, 863–870 (2001).

REVIEWERS' COMMENTS

Reviewer #1 (Remarks to the Author):

The authors have addressed all of the concerns of the previous review. The manuscript is substantially improved and is now suitable for publication.

Reviewer #2 (Remarks to the Author):

The authors have addressed my comments which were minor in the first place. The paper should be accepted for publication.

Reviewer #3 (Remarks to the Author):

The authors were very responsive to the critique and did extensive revisions of their manuscript. They have addressed all my concerns and the paper is now ready for publication.